# Charting γ-secretase substrates by explainable AI

Stephan Breimann [1,2,3,9], Frits Kamp [1,9], Gabriele Basset[1], Claudia Abou-Ajram[1], Gökhan Güner[2,4], Kanta Yanagida[5,6], Masayasu Okochi[6], Stephan A. Müller [2,4], Stefan F. Lichtenthaler [2,4,7], Dieter Langosch [8], Dmitrij Frishman[3] ✉ & Harald Steiner [1,2] ✉

Proteases recognize substrates by decoding sequence information—an essential cellular process elusive when recognition motifs are absent. Here, we unravel this problem for γ-secretase, an intramembrane-cleaving protease associated with Alzheimer's disease and cancer, by developing Comparative Physicochemical Profiling (CPP), a sequence-based algorithm for identifying interpretable physicochemical features. We show that CPP deciphers a γ-secretase substrate signature with single-residue resolution, which can explain the conformational transitions observed in substrates upon γ-secretase binding. Using machine learning, we predict the entire human γ-secretase substrate scope, revealing numerous previously unknown substrates. Our approach outperforms state-of-the-art protein language models, improving prediction accuracy from 60% to 90%, and achieves an 88% success rate in experimental validation. Building on these advancements, we identify pathways and diseases not linked before to γ-secretase. Generally, CPP decodes physicochemical signatures—a concept that extends beyond sequence motifs. We anticipate that our approach will be broadly applicable to diverse molecular recognition processes.

Intramembrane proteases are an important class of proteases, distinctive for their membrane-embedded catalytic residues[1]. Although their substrate cleavage occurs in a relatively site-specific manner[1], consensus cleavage site motifs have not been conclusively identified[2–5]. One of the best-studied intramembrane proteases is γ-secretase, which cleaves about 150 type I single-span membrane proteins within their transmembrane domain (TMD)[6], including the NOTCH receptors, which are implicated in cancer, and the Alzheimer's disease-associated amyloid precursor protein (APP). As shown for APP,

γ-secretase cleaves the TMD initially close to the cytoplasmic border and then trims it progressively[7]. γ-Secretase is critically involved in many important cellular processes, including intracellular signaling and membrane protein homeostasis[8]. However, γ-secretase is thought to cleave only a fraction of the N-out proteome[9] (i.e., all single-span membrane proteins with an extracellular N-terminal ectodomain), and its complete substrate repertoire is not known.

Substrates of γ-secretase must have a short N-terminal ectodomain, typically generated by cleavage of the full-length protein by

[1]Biomedical Center (BMC), Division of Metabolic Biochemistry, Faculty of Medicine, LMU Munich, München, Germany. [2]German Center for Neurodegenerative Diseases (DZNE), DZNE Munich, München, Germany. [3]Department of Bioinformatics, Technical University of Munich (TUM), Freising, Germany. [4]Neuroproteomics, School of Medicine and Health, TUM University Hospital, München, Germany. [5]Department of Pharmacotherapeutics II, Faculty of Pharmacy, Osaka Medical and Pharmaceutical University, Takatsuki, Japan. [6]Neuropsychiatry, Department of Integrated Medicine, Division of Internal Medicine, Osaka University Graduate School of Medicine, Suita, Japan. [7]Munich Cluster for Systems Neurology (SyNergy), SyNergy, München, Germany. [8]Biopolymer Chemistry, TUM, Freising, Germany. [9]These authors contributed equally: Stephan Breimann, Frits Kamp. ✉e-mail: dmitrij.frishman@tum.de; harald.steiner@med.uni-muenchen.de

shedding enzymes ("sheddases"), such as α- and β-secretase (Fig. 1a). Additional factors for γ-secretase substrate recognition and cleavage include TMD backbone flexibility[10–12] (e.g., mediated by glycine- or alanine-based hinges[11,13]) and cooperating N- and C-terminal TMD segments[14], as well as local interactions with γ-secretase exosites[15], tight binding of the TMD to the enzyme[16], and fitting into the active site of the enzyme[17]. However, the relative contributions of these factors for different substrates are still unclear, and they are not all obvious from a substrate sequence. Thus, the molecular principles defining how γ-secretase discriminates substrates from non-substrates (i.e., single-span N-out proteins that are not cleaved) in recruitment and cleavage remain largely unresolved[18–20].

While substrates of many soluble proteases can be reasonably well predicted based on structural properties or sequence patterns[21–24], conserved cleavage motifs have not been identified for γ-secretase substrates (Fig. 1b–d). Considering the substantial number of known γ-secretase substrates, the application of machine learning approaches for substrate prediction is becoming increasingly feasible. Moreover, alignment-free deep learning-based protein language models[25,26], pre-trained on billions of sequences, can also be utilized for prediction tasks with much smaller datasets by transfer learning[27]. However, a key obstacle in applying machine learning to γ-secretase substrate prediction is the lack of negative training data, with only about 15 experimentally validated non-substrates (Supplementary Data 2, Methods "Datasets").

Here, we address the long-standing problem of how γ-secretase recognizes its substrates using a computational workflow (Fig. 2a) based on Comparative Physicochemical Profiling (CPP). This new algorithm was used to identify common physicochemical features of γ-secretase substrates, which were utilized for machine learning-based classification of substrates and non-substrates. To tackle the issue of an imbalanced dataset containing more substrates than non-substrates, we also developed a novel deterministic positive-unlabeled (PU)[28,29] learning approach (dPULearn). The lack of interpretability inherent to machine learning models[30,31] was solved by combining CPP with the artificial intelligence (AI) framework SHapley Additive exPlanations (SHAP)[32], thereby explaining the residue-specific impact of substrate-defining features beyond mere sequence motifs. We achieved a high prediction accuracy for proteins with known substrate status, supported by a similar accuracy in the experimental validation of several substrate and non-substrate candidates.

## Results

### Feature engineering using CPP

CPP is an interpretable feature engineering algorithm that compares two sets of protein sequences to delineate their most discriminative physicochemical features. The core idea of CPP is its feature concept ("CPP feature"), defined as a combination of a "part", a "split", and a "scale" (Supplementary Fig. 1a, Methods "Idea of the CPP algorithm"). Sequence parts—in our case, the TMD and its adjacent N- and C-terminal juxtamembrane domains (JMD-N and JMD-C, respectively) of single-span membrane proteins—can be split into either continuous segments or discontinuous patterns (Supplementary Fig. 1b–d, Methods "Splitting of sequence parts"). These patterns reflect helical periodicity, where residues spaced 3 or 4 positions apart align on the same side of an α-helix[33]. For each resulting split, feature values are computed by assigning to each residue a value of a min-max normalized physicochemical scale (e.g., charge or volume) and averaging them. Scales were obtained mainly from the AAindex database[34] and classified into categories (e.g., conformation) and subcategories (e.g., β-strand), as provided by AAontology[35] (Methods "Classification of scales"). A redundancy-reduced set of 133 scales was obtained using AAclust[36] (Supplementary Fig. 1e, Methods "Selection of scales").

We used CPP to identify features characteristic of γ-secretase substrates. To account for the inherent complexity of defining the exact locations of membrane boundaries, we considered three different sources of TMD annotations (UniProt[37], TMHMM[38], and Phobius[39]; Supplementary Fig. 1f); if not stated otherwise, the TMHMM annotation was used as the default. We compared an expert-curated subset of known γ-secretase substrates (SUBEXPERT, $n = 63$) with a non-redundant reference set of single-span type I transmembrane proteins with unknown substrate status (OTHERS, $n = 631$; Methods "Datasets"). By combining parts, splits, and scales, CPP created over 100,000 features and performed statistical filtering (Supplementary Fig. 1g, Methods "CPP algorithm"), yielding 150 non-redundant features (Supplementary Data 8). These features embody physicochemical properties most discriminative between SUBEXPERT and OTHERS, such as the formation of extended conformations within the TMD-C-JMD-C, as illustrated for APP and the known non-substrate TMX3 in Supplementary Fig. 2a–c.

### Identification of additional non-substrates by dPULearn

To overcome the challenge of robust machine learning posed by an imbalanced dataset comprising 63 substrates (SUBEXPERT) and 14

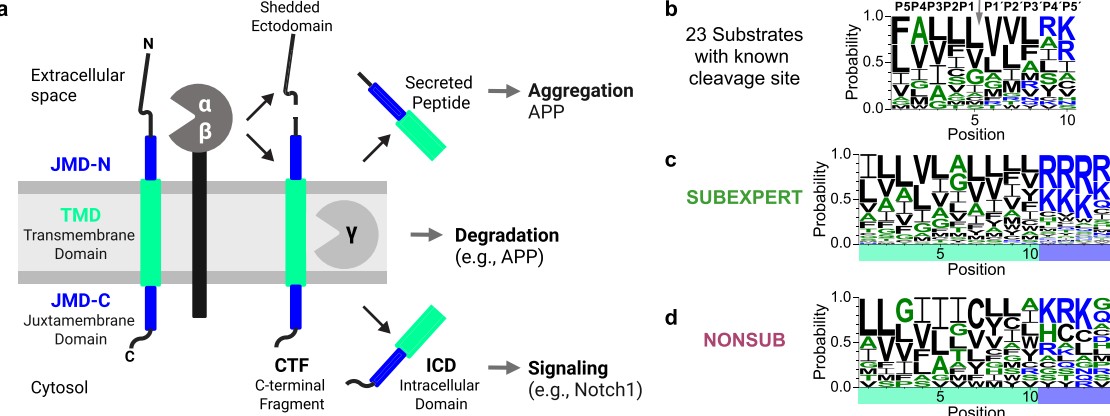

**Fig. 1 | Sequence analysis of the γ-secretase substrate cleavage region.**
**a** Overview of γ-secretase cleavage and functions (Created in BioRender. Breimann, S. (2025) https://BioRender.com/bzswlhf). **b** Sequence logo of the cleavage region of 23 substrates with known cleavage sites. Shown are five positions on the N-terminal (P1–P5) and C-terminal side (P1′–P5′) of the initial cleavage site indicated by ↓. Amino acids are colored according to their physicochemical properties: hydrophobic (black), neutral (green), hydrophilic (blue). **c, d** Sequence logos of the last ten amino acids of the TMD-C and the first four amino acids of the JMD-C for the SUBEXPERT (**c**) and NONSUB (**d**) datasets. See Methods ("Sequence parts of transmembrane proteins", "Datasets", and "Sequence logos") for further details. Source data are provided as a Source Data file.

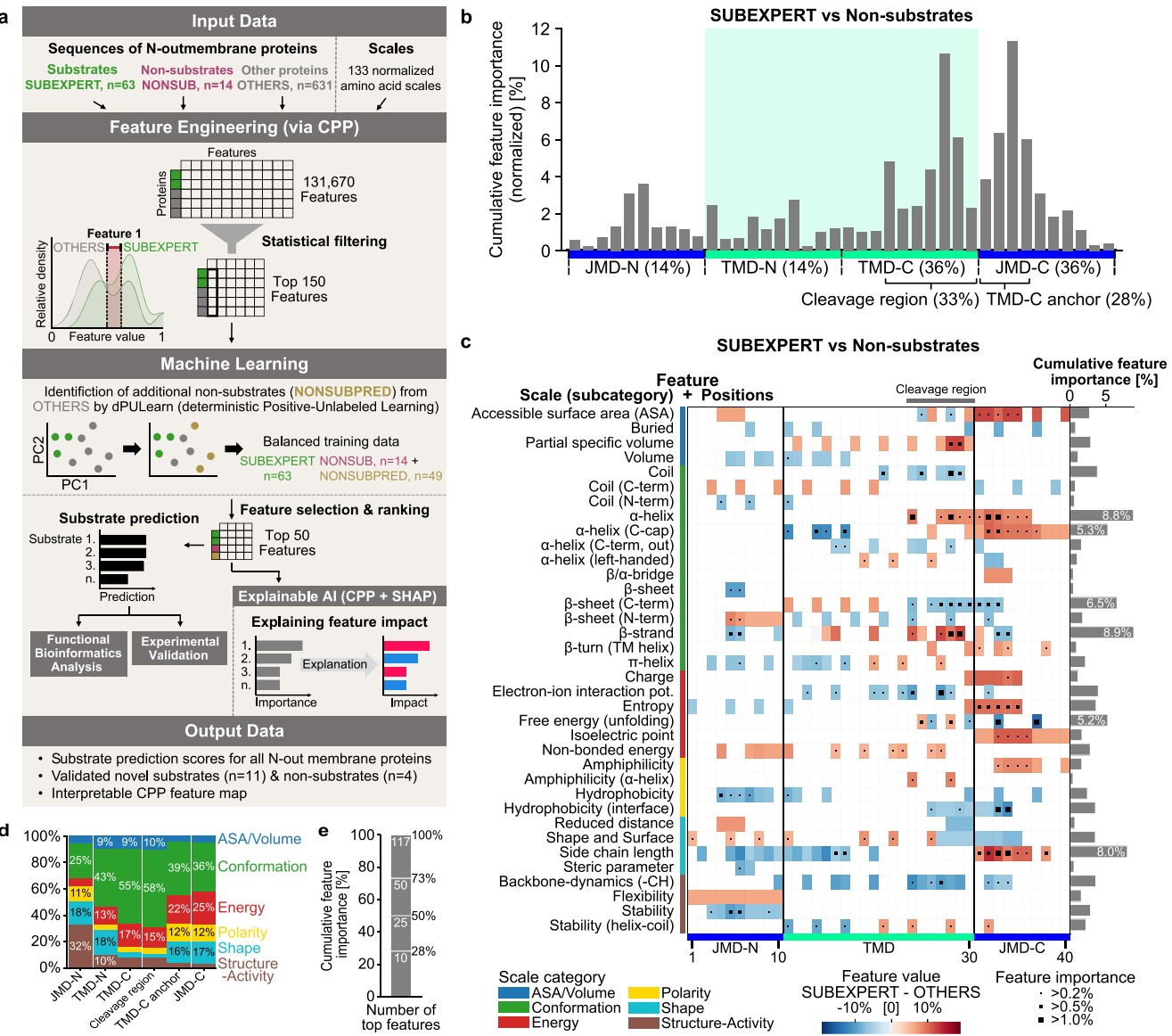

**Fig. 2 | Identification of the physicochemical signature of γ-secretase substrates using CPP.** **a** Workflow comprising the identification of substrate features by Comparative Physicochemical Profiling (CPP), the prediction of substrate candidates using machine learning, and the explanation of feature impacts on substrate prediction scores via Shapley Additive exPlanations (SHAP). **b–e** Results of CPP analysis comparing SUBEXPERT with OTHERS (dataset 1 with TMHMM annotation). Feature importance was obtained by machine learning models trained on SUBEXPERT against non-substrates (NONSUB with NONSUBPRED). Sequence length was set to 40 residues. **b** CPP profile showing cumulative feature importance per residue. Different sequence regions are indicated, including their total feature importance. **c** CPP feature map showing the feature value mean differences (SUBEXPERT - OTHERS) per residue position and scale subcategory, classified into 6 categories as provided by AAontology[35]. The cumulative feature importance per scale subcategory is indicated by gray bars (right). The feature importance per residue position and scale subcategory is highlighted by black squares if higher than 0.2%. **d** Relative occurrence of scale categories per sequence region as shown in (**b**). **e** Cumulative feature importance for top 10, 25, 50, and 117 out of 150 features. Source data are provided as a Source Data file.

non-substrates (NONSUB), we developed a deterministic PU learning algorithm (dPULearn) for identifying additional negatives from unlabeled data based on CPP features (Supplementary Fig. 3a). dPULearn uses principal component analysis to compress the entire feature space (i.e., an $n \times m$ matrix, where $n$ is the number of proteins and $m = 150$ is the number of features) onto principal components (PCs). For each PC, proteins from OTHERS that are most distant from SUBEXPERT proteins are identified as additional non-substrates, based on the absolute distance between their PC value and the mean PC value of SUBEXPERT proteins (Methods "Computational non-substrate identification by dPULearn"). Using dPULearn, we extended the set of 14 known non-substrates by 49 predicted non-substrates (NONSUBPRED), balancing the dataset at 63 substrates and 63 non-substrates (Supplementary Fig. 3b–e).

We benchmarked dPULearn (Supplementary Methods "Benchmarking dPULearn") against the popular PU learning framework developed by ref. [40] (referred to as "Elkanoto"), which uses machine learning classification models and is, therefore, a stochastic approach. Sets of non-substrates generated by the Elkanoto framework and dPULearn were used to assess the prediction performance of two machine learning model types, support vector machine and random forest, as recommended for small datasets[41]. Both approaches showed similar performance (Supplementary Fig. 4a). However, the Elkanoto framework lacked reproducibility (Supplementary Fig. 4b) and consistency (Supplementary Fig. 4c, d). In contrast, dPULearn achieved 100% reproducible results (Supplementary Fig. 4b), high consistency in terms of robustness for selected model hyperparameters (Supplementary Fig. 4e), and a significantly better performance than random

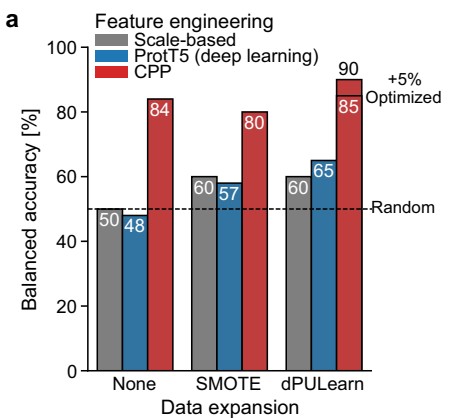

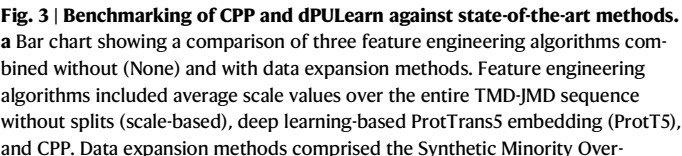

**Fig. 3 | Benchmarking of CPP and dPULearn against state-of-the-art methods.**
**a** Bar chart showing a comparison of three feature engineering algorithms combined without (None) and with data expansion methods. Feature engineering algorithms included average scale values over the entire TMD-JMD sequence without splits (scale-based), deep learning-based ProtTrans5 embedding (ProtT5), and CPP. Data expansion methods comprised the Synthetic Minority Over-sampling Technique (SMOTE) and deterministic Positive-Unlabeled (PU) Learning (dPULearn). Support vector machine models with leave-one-out cross-validation were used for validation. **b** Heatmap showing optimization of the number of CPP features used for model training and non-substrate identification by dPULearn. The optimized result is indicated by a bold square. Source data are provided as a Source Data file.

sampling of non-substrates ($P < 0.001$, two-sided one-sample $t$-test, Supplementary Fig. 4f).

We tested CPP for sets of parts and scales using machine learning models trained on SUBEXPERT vs NONSUB with balanced accuracy (Methods "Machine learning models") as an evaluation measure for predicting the substrate status of these proteins. Without expanding the non-substrates by NONSUBPRED (Supplementary Fig. 4g, Methods "Derivation of the optimal scale and part sets for CPP"), average scale values over the entire sequence of the TMD and its JMDs (TMD-JMD) achieved a balanced accuracy of only 50% when splitting was omitted (referred to as "scale-based" feature engineering). When applying splitting on either the TMD alone or the TMD-JMD, CPP performed similarly. However, CPP achieved 84% balanced accuracy with an optimized part set and 92% accuracy when NONSUBPRED was included to balance the datasets (Supplementary Fig. 4g, h, Supplementary Methods "Benchmarking CPP without and with NONSUBPRED").

## Feature ranking using machine learning

To rank the features obtained by CPP, machine learning models were trained for the three TMD annotations and on two alternative training datasets (Supplementary Fig. 5a, b, Methods "Training datasets") using as positive training data either expert-curated substrates (SUBEXPERT, $n = 63$) only (dataset 1) or both expert-curated and literature-based[6] substrates (SUBEXPERT and SUBLIT, $n = 136 = 63 + 73$) (dataset 2). In both datasets, the same negative training data (NONSUB and NONSUBPRED, $n = 63 = 14 + 49$) was included. For each annotation-dataset combination, 10 different machine learning classification model types —four tree-based, two linear, one kernel-based, one neural network, and two ensemble model classes—were used in 25 training rounds, yielding $250 = 25 \times 10$ trained models.

Each training round involved data splitting, recursive feature selection, model hyperparameter optimization, and substrate prediction (Methods "Learning strategy"). The number of pre-selected features was optimized at every round, and the results of all models were aggregated for model evaluation and substrate prediction (Supplementary Fig. 5c, d). For feature ranking, the feature importance was obtained directly from the four tree-based model types ($100 = 25 \times 4$ trained models) and averaged. If not stated otherwise, results are described for dataset 1 with TMHMM annotation, as this has the best performance (Supplementary Fig. 5d). These ranked features form the basis for distinguishing γ-secretase substrates from non-substrates.

## Physicochemical signature of γ-secretase substrates

To enable an insightful interpretation of the CPP features and their importance as obtained by machine learning models (Fig. 2a), we developed the "CPP profile" and "CPP feature map" visualizations (Fig. 2b, c). The CPP profile shows the cumulative feature importance per amino acid position within the different TMD-JMD sequence regions (Fig. 2b). Remarkably, for all identified 150 features, two regions exhibit the highest cumulative feature importance: the region around the initial γ-secretase cleavage site within the TMD-C (33%) and the first four residues of the JMD-C, referred to as "TMD-C anchor" (28%; Fig. 2b, Methods "Sequence parts of transmembrane proteins").

Further detail for this analysis is offered by the CPP feature map (Fig. 2c), which illustrates the mean differences of feature values between SUBEXPERT and OTHERS (red/blue indicates higher/lower feature values for substrates) per residue position and for each scale subcategory from AAontology[35] (color-coded, left), combined with feature importance. This map reveals that among the most important properties discriminating substrates from non-substrates (highlighted by black squares) are: (a) within the TMD-C anchor, residues with increased helix termination propensity ("α-helix (C-cap)"), large "side-chain length" and "accessible surface area (ASA)", as well as increased "charge" and disorder ("entropy"); (b) at the TMD-C/JMD-C interface, residues with an increased α-helical ("α-helix") and decreased β-sheet termination ("β-sheet (C-term)") tendency; and (c) around the initial cleavage site within the TMD-C, an increased conformational preference for both extended ("β-strand") and helical ("α-helix") structure as well as altered unfolding propensities ("free energy (unfolding)") and reduced "backbone dynamics (-CH)". In addition, substrates are characterized by small residues (e.g., reduced "side-chain length") within the TMD-N and flexibility-inducing residues (e.g., reduced "stability") within the JMD-N.

Overall, conformational features are dominant (Fig. 2d). Of the 150 CPP features, the ten most important ones constitute 28% of the cumulative feature importance, the top 50 account for 73%, and the top 117 for 100% (Fig. 2e), indicating that the last 33 features contribute minimally to γ-secretase substrate identification. We refer to this set of CPP features and their importance as the common physicochemical signature of γ-secretase substrates.

## CPP and dPULearn outperform state-of-the-art methods

To evaluate how CPP and dPULearn perform against state-of-the-art protein prediction methods, we compared them with both a scale-

based and a deep learning-based feature engineering approach, combined with the Synthetic Minority Over-sampling Technique (SMOTE)[42,43] for handling imbalanced datasets. Scale-based features were generated by averaging physicochemical properties (e.g., polarity, charge, volume) across the entire TMD-JMD sequence of a protein, creating for each scale a single representative value used as a feature. For the deep learning-based approach, we used the ProtTrans5 ('ProtT5') language model[25,26], which produced numerical vectors, known as embeddings, for each protein sequence. These protein embeddings represent scale-like residue properties learned by ProtT5 from large protein sequence datasets. To ensure comparability between the scale-based and embedding-based approaches, we also averaged the embedding values across the entire TMD-JMD sequence.

For each combination of feature engineering methods (scale-based, embeddings, CPP) and data expansion techniques (None, SMOTE, dPULearn), support vector machine models with default settings were trained and consistently evaluated by leave-one-out cross-validation on SUBEXPERT and NONSUB (Supplementary Methods "Evaluation measures") to compare the different approaches in a standardized baseline machine learning setting. Support vector machine models employing scale-based feature engineering or embeddings showed only ~50% balanced accuracy without data expansion and ~60% with SMOTE (Fig. 3a). Models employing embeddings and dPULearn reached 65%. In contrast, models using CPP features achieved a balanced accuracy of 84% without data expansion and up to 90% with dPULearn when optimized by testing different numbers of CPP features for model training and non-substrate identification with dPULearn (Fig. 3b, Supplementary Methods "Benchmarking CPP and dPULearn against deep learning-based embeddings").

## Substrate prediction using machine learning

The probability of a given protein being cleaved by γ-secretase (termed "substrate prediction score") was computed as the average prediction score over the six best-performing dataset-annotation combinations, using a total of $1500 = 250 \times 6$ trained machine learning models (Supplementary Fig. 5d–f, Methods "Aggregation of prediction results"). As expected, the substrate prediction scores differed between substrate and non-substrate datasets (Fig. 4a), showing, with a few exceptions, the highest scores for proteins from SUBEXPERT (>80%), scores between 50 and 95% for SUBLIT, and scores <50% for NONSUB and NONSUBPRED. The substrate prediction scores for proteins from our reference set (OTHERS) ranged between 10 and 95% (Supplementary Fig. 5e). When considering dataset-annotation combinations independently (Supplementary Fig. 5f), lower prediction scores were obtained for dataset 1, yielding a right-skewed distribution of prediction scores for OTHERS, and vice versa for dataset 2. This dataset-dependent effect on substrate prediction scores became even more pronounced when they were averaged over the three TMD annotation sources separately for datasets 1 and 2 (Supplementary Fig. 6a). The consistency of the prediction scores was higher within TMD annotations (e.g., Pearson's $r = 0.9$ for dataset 1 vs dataset 2, TMHMM) than within datasets (e.g., Pearson's $r = 0.74$ for UniProt vs TMHMM, dataset 1) (Supplementary Fig. 6b).

Very low or high machine learning prediction scores generally indicate higher model confidence. In our approach, extreme substrate prediction scores for a given protein reflect consistent predictions across the six dataset-annotation combinations. We defined the following confidence-based substrate classes encompassing the continuum of γ-secretase substrate prediction scores: high-confidence (HC) non-substrates (0–20%), low-confidence (LC) non-substrates (>20–50%), LC substrates (>50–80%), and HC substrates (>80–100%) (Fig. 4b, Methods "Confidence-based substrate classes"). The 80% HC substrate cut-off was chosen because with this threshold ~90% of the SUBEXPERT proteins and 0% of proteins from the non-substrate

datasets are classified as HC substrate (Fig. 4c). Compared to HC classes, substrate prediction scores for LC classes had higher standard deviations (Fig. 4d–f), supporting our threshold choices. Notably, all proteins from SUBEXPERT score over 50%, while 15 proteins from SUBLIT score below this LC substrate cut-off. However, these outliers are all classified as LC non-substrates, reflecting the uncertainty of the LC classes. Among these proteins is the Alzheimer's disease-associated TREM2, which is an uncommon substrate due to a charged lysine residue in its TMD[44]. Charting the complete human single-span N-out proteome ($n = 1534$; Methods "Datasets") by substrate prediction scores reveals that 250 proteins belong to the HC substrate class (Fig. 4b, see Supplementary Data 13 for the prediction results), while 98, 599, and 587 proteins were classified as HC non-substrates, LC non-substrates, and LC substrates, respectively.

## Experimental validation of predicted substrates and non-substrates

We experimentally validated predicted γ-secretase substrates using established cell-based cleavage assays (Supplementary Fig. 7a). Candidates were chosen across all four confidence classes (Supplementary Data 14). For the selection of biologically interesting HC substrates (Methods 'Selection of substrate and non-substrate candidates'), we devised a "relevance score" [0–1] (Fig. 4g) based on five equally weighted factors such as associations with diseases or pathways that have not been previously linked to γ-secretase or its known substrates (Methods "Computation of relevance score").

As an experimental readout for the substrate status, we analyzed substrate accumulation following expression of C-terminally 10×His-tagged candidate proteins in HEK293 cells with or without a double knockout (DKO) of the catalytic subunits of γ-secretase, presenilin 1 (PS1) and presenilin 2 (PS2)[45] (Methods "Cell-based cleavage assays"). As shown in Fig. 5a, compared to control cells without DKO, the C-terminal fragments (CTFs) of five HC and six LC predicted substrate candidates accumulated in the DKO cells, confirming them as substrates of γ-secretase. Validated HC substrate candidates are CD2, CD68, CD86, ERBB2, and FAM174A. Notably, the immune system-related CD68 and CD86, as well as the cancer-related ERBB2, have a relevance score of 0.8 (Fig. 4g). We also validated the LC substrate candidates CLMP, ICAM1, PCDH17, as well as the cancer-related GPNMB and TIMD4 as substrates. While shedding was previously only reported for ERBB2, GPNMB, ICAM1, and TIMD4 (Supplementary Data 3, 4, 14), our results imply the existence of a sheddase for the other validated substrates. The LC substrate tyrosine-protein kinase STYK1 was validated by the accumulation of its full-length protein because shedding is not required due to its naturally short ectodomain (i.e., containing less than 75 residues)[46,47]. Additionally, the substrate status of all candidates was confirmed by pharmacological γ-secretase inhibition (Supplementary Fig. 7b). VCAM1, predicted as an LC non-substrate with a substrate prediction score of 32 ± 16%, was determined to actually be a substrate, thus being the only positive outlier.

For non-substrate candidates, we selected proteins with a naturally short ectodomain of ~30 amino acids to circumvent the initial shedding requirement, thereby facilitating unambiguous validation. Accordingly, we validated one HC non-substrate (ACSL5) and three LC non-substrates (FAAH2, MANBAL, SLC27A1; Fig. 5b and Supplementary Fig. 7c) by the unchanged levels of the full-length protein. One negative outlier was found, the predicted LC substrate RELL2 (53 ± 21%).

We next asked whether the main features identified by CPP (Fig. 2b, c) are sufficient to define substrate cleavage. To this end, we analyzed the cleavability of selected biotinylated peptides designed to solely encompass their TMD and C-terminal flanking region[48] in a cell-free γ-secretase assay (Supplementary Fig. 8a–c, Methods "Cell-free cleavage assays"). Immunoblot analysis indeed showed that the APP-, APLP2-, NOTCH1-, NOTCH2-, and ERBB2-based peptide substrates were specifically cleaved by γ-secretase (Fig. 5c), whereas the non-

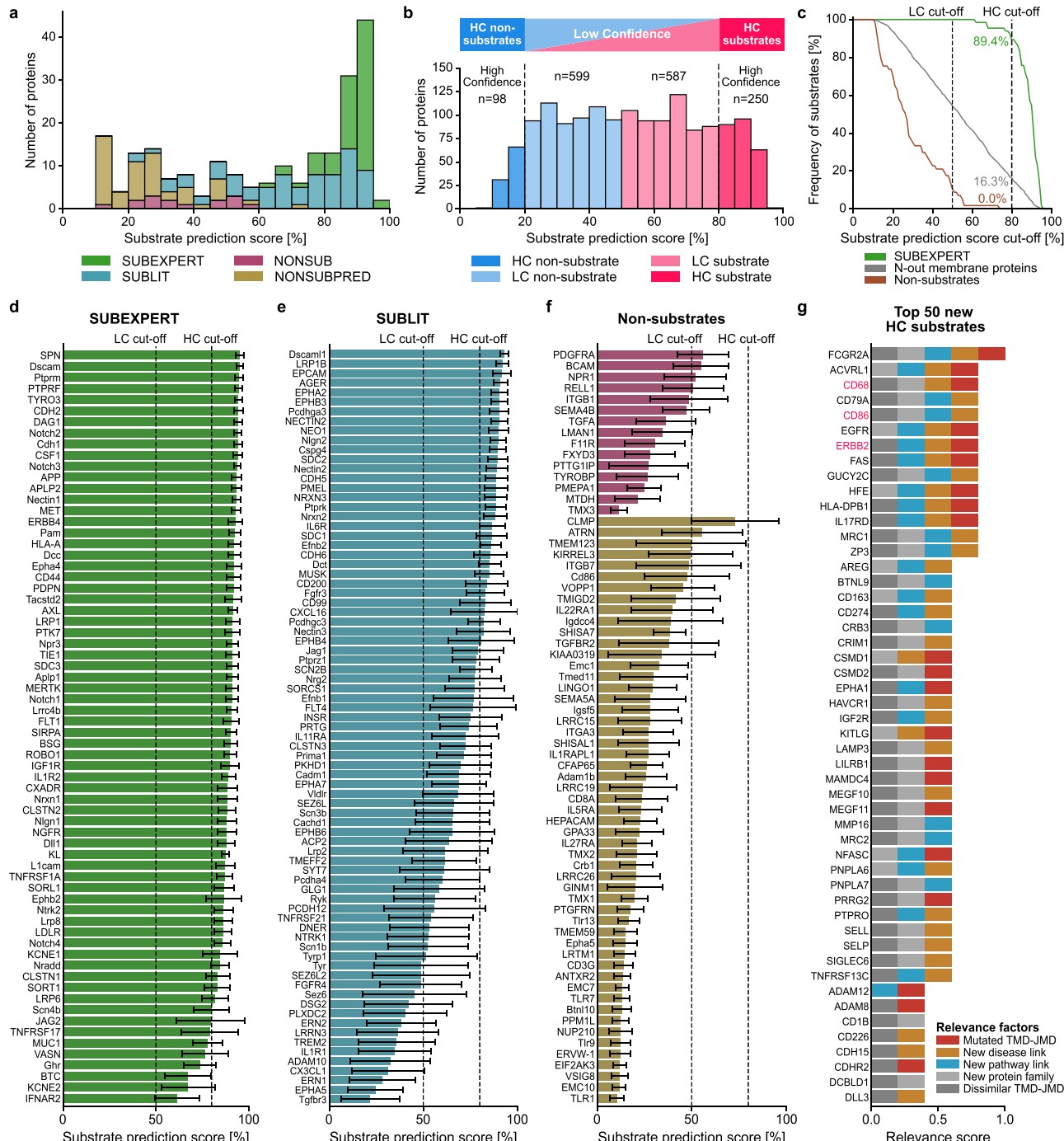

**Fig. 4 | Prediction of γ-secretase substrates and non-substrates. a** Stacked bar chart showing the number of proteins against the substrate prediction score for expert-curated γ-secretase substrates (SUBEXPERT, green), other published substrates (SUBLIT, blue), established non-substrates (NONSUB, purple), and non-substrates predicted by dPULearn (NONSUBPRED, brown). **b** Bar chart showing the number of proteins against substrate prediction score for all human N-out single-span membrane proteins, categorized as high-confidence (HC) or low-confidence (LC) substrates (red color tones) and non-substrates (blue color tones). **c** Percentage of SUBEXPERT proteins (green), non-substrates (NONSUB + NONSUBPRED, brown) and all N-out membrane proteins (gray) as a function of the substrate prediction cut-off. **d–f** Substrate prediction scores for the UniProt annotation-based datasets of **d** SUBEXPERT (*n* = 68), **e** SUBLIT (*n* = 79), and

**f** NONSUB (*n* = 15) and NONSUBPRED (*n* = 53), given here instead of the TMHMM default to show all known substrates and non-substrates (see Supplementary Fig. 1f). Color code as in (**a**). In **a–f**, substrate prediction scores are shown aggregated over the six best approaches for each dataset-annotation combination (see Methods "Aggregation of prediction results"). In **d–f**, the substrate prediction scores are shown as mean ± standard deviation, reflecting variability across approaches and overall prediction uncertainty. **g** Bar chart showing relevance score for the 50 new HC substrates with the highest substrate prediction score. Experimentally validated substrates are highlighted in red. In **d–g**, gene names are in uppercase for human and the first letter capitalized for other organisms (Supplementary Methods "Datasets"). Source data are provided as a Source Data file.

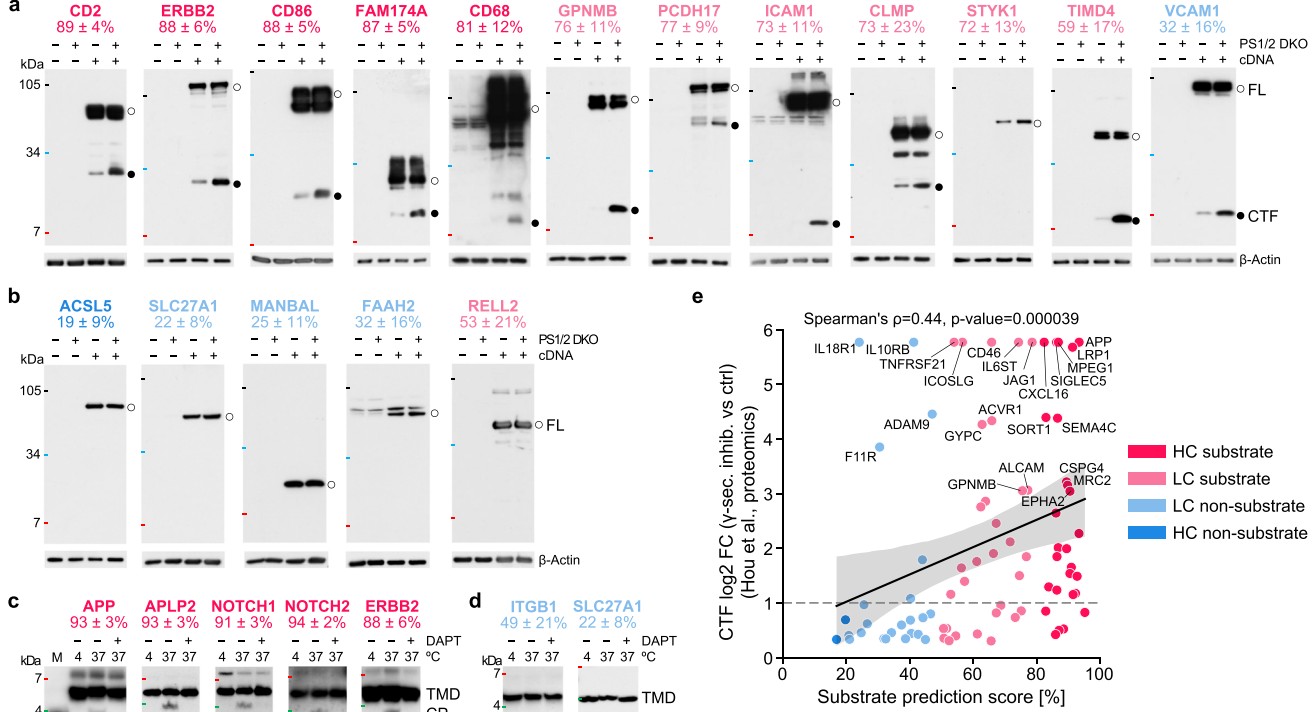

**Fig. 5 | Experimental validation of predicted substrates and non-substrates.**
**a** Immunoblot analysis of substrate candidates tested in a cell-based cleavage assay.
C-terminal fragment (CTF, ●) accumulation in PS1/2 DKO cells validates their
substrate status. For STYK1, the full-length (FL) protein form (○) accumulated as
this protein can be cleaved directly by γ-secretase. For ERBB2, a smaller band than
the expected molecular weight (MW) of its CTF was found, likely due to subsequent
caspase cleavage of the accumulating CTF[97]. **b** Immunoblot analysis of non-
substrate candidates, performed as in (**a**). Unchanged FL levels validate their non-
substrate status. In **a**, **b**, β-Actin served as a loading control. **c**, **d** Immunoblot
analysis of substrate and non-substrate TMD-based peptides (TMD) tested in a cell-
free cleavage assay. Cleavage products (CP) were found after incubation at 37 °C for
substrate peptides (**c**), but not non-substrate peptides (**d**). To control for cleavage
specificity, the γ-secretase inhibitor (GSI) DAPT was used[81]. For APP, LVMLKKK-
Biotin was used as a CP migration standard (M) in (**c**). Substrates and non-substrates
are indicated by their gene name followed by their substrate prediction scores ±

standard deviation (see Methods "Aggregation of prediction results") using the
color code for confidence-based substrate classes (Fig. 3b). VCAM1 and RELL2 are
outliers in that they proved to be substrate or non-substrate, respectively, contrary
to the prediction. Small black, blue, and red lines indicate 105, 34, and 7 kDa MW
markers. All tested candidates were from human, except CD68, ICAM1, STYK1,
ACLS5, and SLC27A1, which were from mouse (see Methods "Selection of substrate
and non-substrate candidates"). Immunoblot analyses in **a**–**d** are representative of
three independent experiments. **e** A scatterplot showing the correlation (two-sided
Spearman correlation) between substrate prediction scores and recently reported
log2 fold change (FC) of CTF accumulation in the presence of a GSI against control,
obtained for 85 endogenously expressed proteins in human microglia-like cells[49].
The maximum FC was used for proteins with CTF detection only during inhibition.
The regression estimate (solid black line) with 95% confidence interval (gray shaded
area) and the chosen substrate identification FC threshold (dashed gray line) are
indicated. Source data are provided as a Source Data file.

substrate peptides of ITGB1[9] and SLC27A1 were not (Fig. 5d). In total,
with a 50% cut-off, 11 substrate candidates (5 HC and 6 LC substrates)
and 4 non-substrate candidates (1 HC and 3 LC non-substrates) were
validated. Only two predictions were incorrect, yielding a success rate
of 88% (15 out of 17), consistent with our computational accuracy
(90%, Fig. 3a).

In line with our results, recent studies identified and biochemically
validated additional γ-secretase substrate candidates for which we
determined substrate prediction scores of ~50% (CD300A, MILR1, and
TNRSF1B[49]), above 70% (TNR12[47]), and above 90% (PTPRD[50] and
PTPRT[51]) (Supplementary Data 13). To assess our substrate predictions
for endogenously expressed γ-secretase substrates, we used a recent
proteomics screen that identified 85 substrate candidates in human
microglia-like cells by pharmacological γ-secretase inhibition[49]. We
observed a significant positive Spearman correlation (ρ = 0.44,
P < 0.001, two-sided) between our substrate prediction scores and
their reported fold changes of CTF accumulation (Fig. 5e). Using a log2
fold change above 1 as identification threshold, 42 out of 48 proteins
were predicted as substrates, yielding a success rate of 88%. These 48
proteins included 28 proteins of unknown substrate status, of which 24
were predicted to be substrates, yielding a similar success rate of 86%.
These success rates are consistent with our experimental results (88%),
both aligning with our computational accuracy of 90% (Fig. 3a).

## Explaining prediction results at the amino acid sequence level
To interpret the substrate prediction scores of individual proteins, we
combined CPP with the explainable AI framework SHAP[32] (Supple-
mentary Fig. 9a, Methods "Explainable AI using SHAP" and "Combining
CPP with SHAP"). SHAP quantifies the contribution of each feature
("feature impact") to increase or decrease prediction scores by positive
and negative SHAP values, respectively. The sum of these additive
values, referred to as the SHAP value sum, ranges from 0 to 1 and
approximates the prediction score [0–100%], as illustrated by SHAP
force plots for APP, ITGB1, and TMX3 (Supplementary Fig. 9b). We
developed four further visualizations: "CPP-SHAP ranking plot", "CPP-
SHAP profile", "CPP heatmap", and "CPP-SHAP heatmap" (Supple-
mentary Methods "CPP-SHAP plots").

To assess CPP and SHAP against conventional sequence similarity
approaches (Fig. 6), we first compared two HC substrates, APP and
NOTCH2, with the LC non-substrate ITGB1 (Fig. 6a–f). Despite their low
TMD-JMD sequence similarity (21%; Fig. 6g, Methods "Comparison of
CPP with a similarity-based approach"), APP and NOTCH2 have nearly
identical CPP-SHAP profiles with almost exclusively positive-impact
features (Fig. 6d, e). Both echo general substrate-defining properties,
characterized by similar CPP features over all annotations (Supple-
mentary Fig. 9c). In contrast, ITGB1, which also has a low TMD-JMD
sequence similarity to APP (19%, Fig. 6g), shows different CPP-SHAP

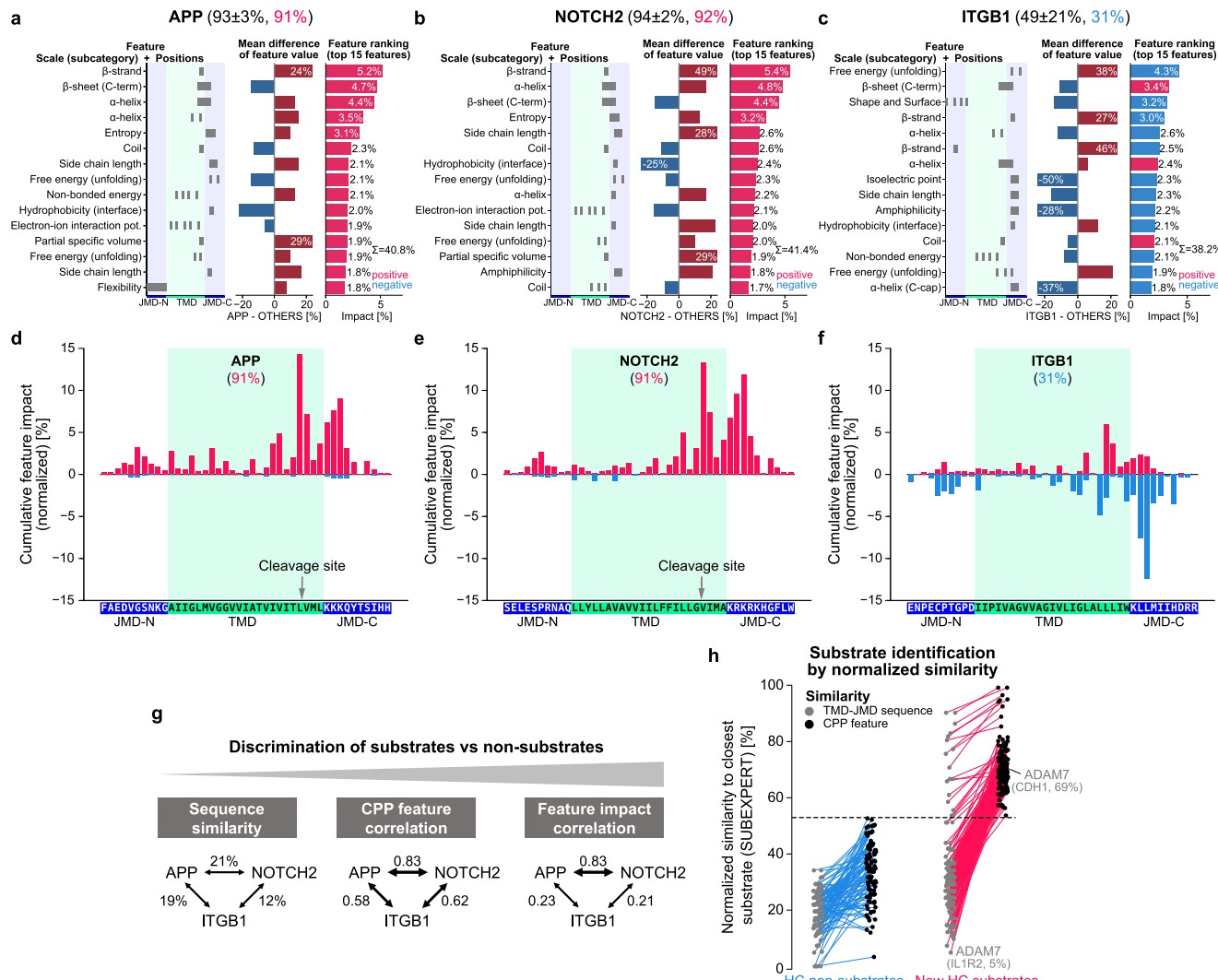

**Fig. 6 | Explainable AI analysis of substrate-defining features for APP, NOTCH2, and ITGB1. a–c** CPP-SHAP ranking plots showing the top 15 features explaining the substrate prediction scores of the high-confidence (HC) substrates APP (**a**) and NOTCH2 (**b**), as well as of the low-confidence (LC) non-substrate ITGB1 (**c**). The substrate prediction scores ± standard deviation (see Methods "Aggregation of prediction results") for APP, NOTCH2 and ITGB1 are given, followed by their color-highlighted prediction scores (based on dataset 1 and TMHMM annotation) explained by SHAP. Indicated are the scale subcategories, residue positions of part-split combinations, differences in the mean feature value (compared to OTHERS), and the feature impact (based on TMHMM, dataset 1 training). Features are ranked according to their positive (blue) or negative (red) impact. Σ indicates the sum of the importance of all top 15 features. **d–f** CPP-SHAP profiles showing the cumulative feature impact per residue for the TMD-JMD sequence of APP (**d**), NOTCH2 (murine

sequence), and ITGB1 (**f**).The feature impact was obtained based on dataset 1 with TMHMM annotation (see Methods "Combining CPP with SHAP"). **g** Comparison of discriminative power (substrates vs non-substrates) for different similarity measures exemplified for APP, NOTCH2, and ITGB1. Arrow thickness corresponds to similarity strength. **h** A scatterplot showing the normalized similarity to the closest (i.e., most similar or correlating) substrate from SUBEXPERT for all HC non-substrates (blue) and new HC substrates (red). A pair of connected dots represents the normalized similarity values for a particular protein based on the TMD-JMD sequence (gray) or CPP feature correlation (black). The closest substrate can differ between both measures, as exemplified by the new HC substrate ADAM7. Min-max normalization was performed on the human N-out proteome dataset. The dashed black line indicates the discrimination border based on CPP features. Source data are provided as a Source Data file.

plots, in which negative-impact features predominate (Fig. 6c, f). To further demonstrate the discriminative power of CPP features, we compared the ability of CPP feature similarity vs TMD-JMD sequence similarity to distinguish HC non-substrates from HC substrates (Fig. 6h, Supplementary Methods "Comparison of CPP with a similarity-based approach"). Both measures were given as min-max normalized similarity to the most similar ("closest") substrate in SUB-EXPERT. For example, the HC substrate candidate ADAM7 has a mere 5% TMD-JMD sequence similarity to IL1R2 but shares a 69% CPP feature similarity with CDH1. Remarkably, while sequence similarity was unsuitable for discrimination, the separation based on CPP features was perfect (Fig. 6h, dashed line).

SHAP values depend on the machine learning model, the dataset, and how samples are labeled during training—i.e., whether they are marked as positive or negative. When extending our SHAP analysis to proteins that are not included in the training dataset, the challenge arises of whether to label them as positive (i.e., substrate) or negative (i.e., non-substrate). To address this labeling ambiguity, we developed fuzzy labeling (Supplementary Fig. 10, Supplementary Methods "Fuzzy labeling"). In this labeling procedure, machine learning models are trained over multiple rounds, and the frequency of labeling proteins with unknown status (as positive or negative) is set corresponding to their prediction score. This probabilistic approach is designed for proteins that are absent from a training

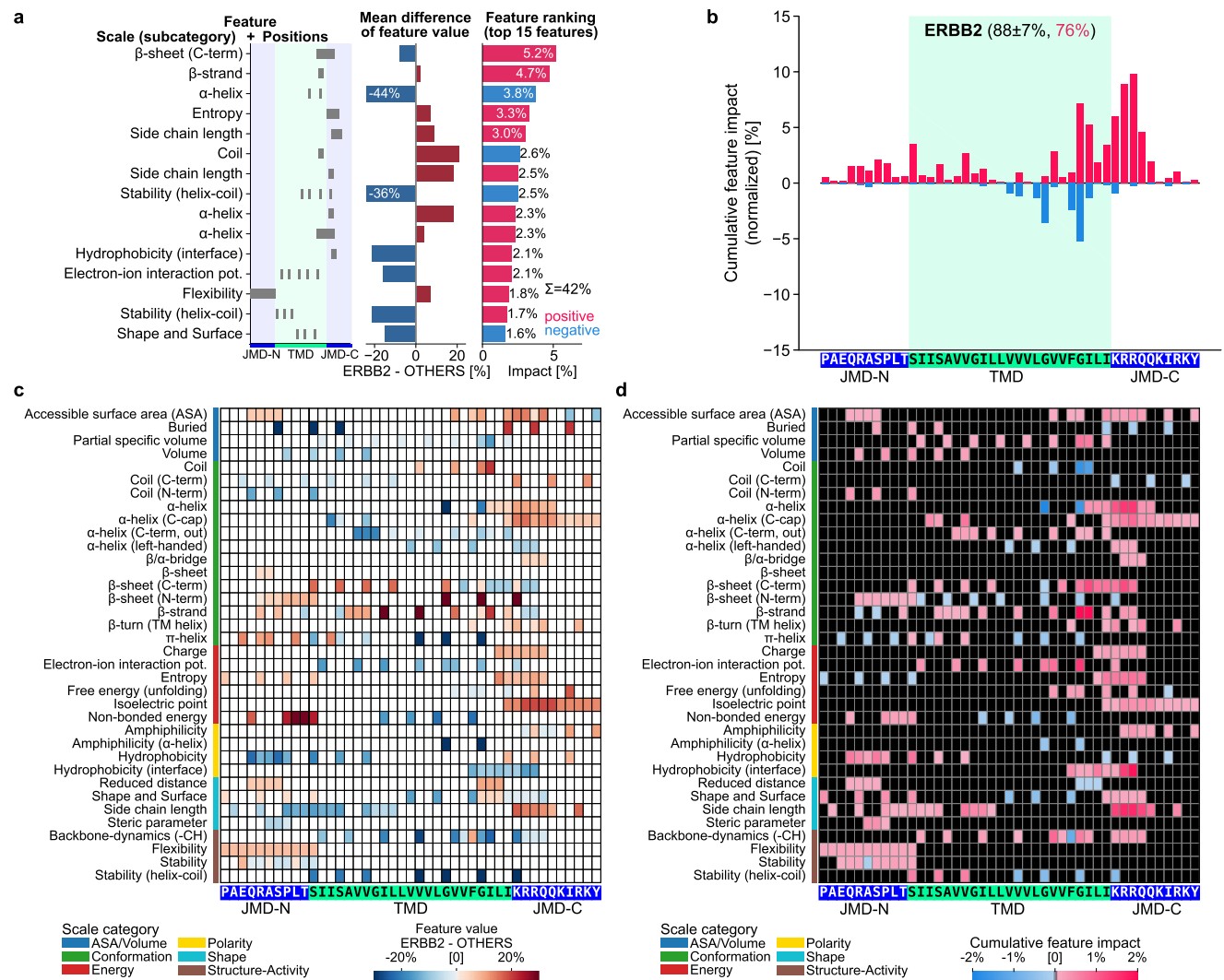

**Fig. 7 | CPP-SHAP analysis for ERBB2.** The validated HC substrate ERBB2 analyzed by four CPP-SHAP plots using fuzzy labeling (see Methods "Combining CPP with SHAP"): The CPP-SHAP ranking plot (**a**) ranks the top 15 features by the absolute value of their impact, which can be positive (red) or negative (blue); the CPP-SHAP profile (**b**) shows the cumulative feature impact per residue; the CPP heatmap (**c**) highlights the differences in feature values between the respective protein and the reference dataset (OTHERS) per scale subcategory and residue; and the CPP-SHAP heatmap (**d**) illustrates the feature impact per scale subcategory and residue. Scale categories are from AAontology[35] and uniformly color-coded. The CPP-SHAP

analysis results for ERBB2 (88 ± 7% substrate prediction score) explain its prediction score of 76% (red) based on dataset 1 with TMHMM annotation. The CPP-SHAP ranking plot shows the predominantly positive impact of the top 15 features, such as an increased β-strand tendency in the TMD-C or an increased entropy in the TMD-C anchor. The positive impact of these regions is underlined in the CPP-SHAP profile. The CPP heatmap and CPP-SHAP heatmap reveal the negative impact of some residues within the TMD-C, such as two glycines, due to their α-helix destabilizing effect. For comparison, see the CPP-SHAP analysis for the LC substrate SLC27A1 (Supplementary Fig. 11). Source data are provided as a Source Data file.

dataset and is particularly useful for explaining uncertain prediction scores reflected by high variance. We showcased fuzzy labeling for TREM2, which is a substrate from SUBLIT, but predicted as a low-confidence non-substrate (36 ± 20%).

In dataset 1, where TREM2 is not included, its prediction score is 22%. However, labeling TREM2 as substrate yields a SHAP value sum of 0.66 (Supplementary Fig. 10a). Applying fuzzy labeling on TREM2 during 25 training rounds (i.e., labeling it 5 rounds as substrate and 20 rounds as non-substrate) results in a SHAP value sum of 0.27, closely reflecting its 22% prediction score (Supplementary Fig. 10b). For dataset 1, the low prediction score of TREM2 is explained by the lack of disordered, large, and charged residues in the TMD-C anchor (Supplementary Fig. 10c, d). For dataset 2, which contains TREM2, it is always labeled as a substrate so that fuzzy labeling is not applied. This results in a higher prediction score of 62%, explained by positive-impact features overruling negative ones (Supplementary Fig. 10e). Basically, fuzzy labeling ensures that SHAP values reliably approximate

prediction scores for any protein with unknown substrate status, which is especially useful for prediction scores with high variance. This probabilistic labeling approach allows an in-depth comparison of the physicochemical signatures of any protein with a predicted substrate status, as illustrated for the validated HC substrate ERBB2 (Fig. 7) and LC non-substrate SLC27A1 (Supplementary Fig. 11).

To discover substrate subgroups based on the feature impact, we hierarchically clustered dataset 1 using Pearson correlation as a similarity measure, yielding five clusters in concordance with our confidence-based substrate continuum (Fig. 8, Supplementary Methods "Clustering based on feature impact"). Cluster 1 contained mainly HC substrates, such as APP or APLP2; cluster 2 comprised not only HC substrates, but also LC substrates and one LC non-substrate (ITGB1); clusters 3 and 4 included LC substrates and non-substrates, reflecting the uncertainty of their prediction. In contrast, cluster 5 was dominated by HC non-substrates such as TMX3. Supplementary Fig. 12 compares the feature impact of four proteins selected from clusters 1, 3, 4, and 5.

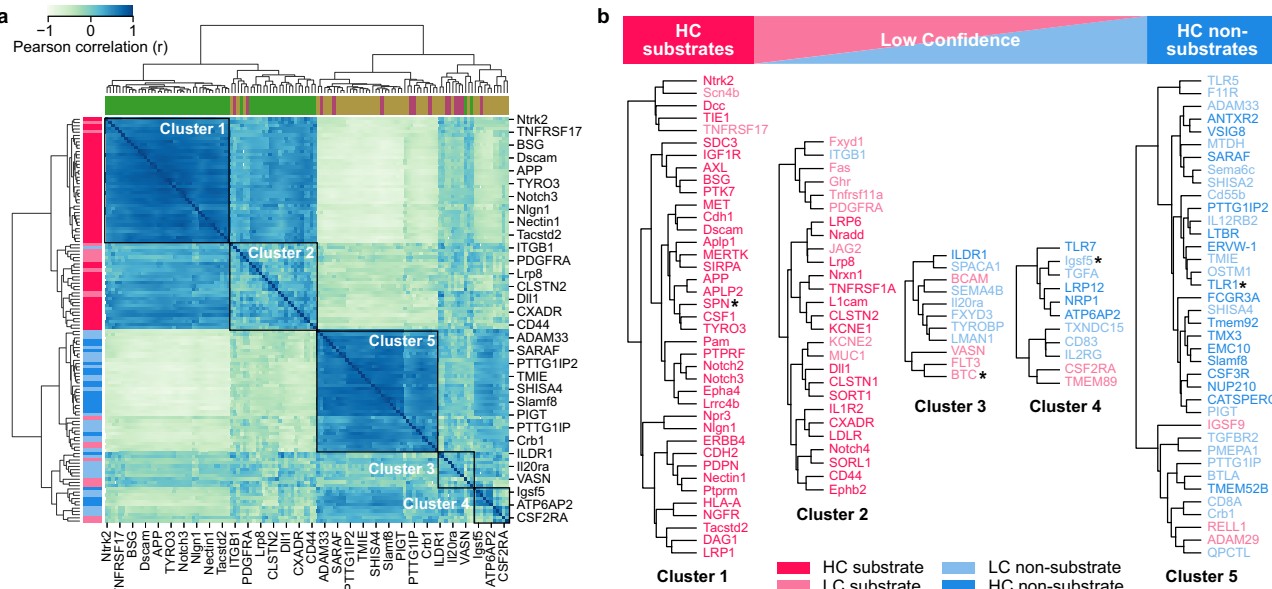

**Fig. 8 | Clustering of confidence-based substrate classes based on feature impact. a** Heatmap showing hierarchical clustering of dataset 1 (with TMHMM annotation) based on Pearson correlation coefficients for feature impacts (see Supplementary Methods "Clustering based on feature impact"). Dataset classes (top, color code of SUBEXPERT, NONSUB, and NONSUBPRED, according to Fig. 3a) and corresponding confidence-based substrate classes (left, color code according to (**b**)) are indicated. Five distinct clusters are highlighted by squares. **b** Five clusters from (**a**) depicted along the continuum of confidence-based substrate classes (top), ranging from high-confidence (HC) substrates, through low-confidence (LC) substrates and LC non-substrates, to HC non-substrates. The color code indicates the confidence-based substrate classes (Methods "Confidence-based substrate classes"). See Supplementary Fig. 12 for the CPP-SHAP analysis results of the four selected proteins highlighted by asterisks (SPN, BTC, murine IGSF5, and TLR1). Gene names are in uppercase for human and with the first letter capitalized for other organisms (Supplementary Methods "Datasets"). Source data are provided as a Source Data file.

## Functional bioinformatics analysis of γ-secretase substrates

To gain deeper insight into the biological role of γ-secretase, we performed a functional bioinformatics analysis of the human N-out proteome ($n = 1534$) (Fig. 9a–c, Supplementary Data 17, and Supplementary Methods "Dataset of human N-out proteome"), focusing on HC substrates ($n = 250$) and their subgroup of HC substrate candidates ("new HC substrates", $n = 160$). HC substrates showed a significant enrichment for Gene Ontology (GO) terms (Methods "Enrichment analysis"), such as cell adhesion and cell periphery (Fig. 9d). We also identified pathway terms only enriched for new HC substrates, including allograft rejection or type I diabetes mellitus; notably, among these were pathway terms not previously associated with γ-secretase or its substrates, such as glycerophospholipid catabolism and class 3 semaphorin (SEMA3A) related pathway terms (Fig. 9e). Clustering all significantly enriched pathway terms yielded 7 clusters (C1–C7) related to functions such as cell communication (C1) or immune regulation (C7) (Fig. 9f). Network analysis (Methods "Network analysis") of the new HC substrates revealed 8 significantly over-represented modules[52] (M1–M8) associated with, for example, immune diseases (M3) or semaphorin interaction (M5) (Supplementary Fig. 13).

To further characterize the new HC substrates of the 8 modules (Supplementary Fig. 14a), we computed their relevance scores, which included a "mutated TMD-JMD" factor accounting for mutations within the TMD-JMD. Several of these mutations are disease-associated, similar to the Alzheimer's disease-causing APP London (V717I) mutation[53]. Subsequently, we selected per module the two proteins with the highest number of pathway links (preferring validated substrates) and integrated modules with clusters (Supplementary Fig. 14b). Notably, the experimentally validated CD86 (M2) had many associations with immune disease terms (C4), such as type I diabetes mellitus or autoimmune thyroid disease.

Next, we obtained all pathway (Reactome), disease (DisGeNET), and mutation links for each protein of the human N-out proteome

(Supplementary Fig. 15a, Methods "Analysis of pathway, disease, and mutation links"). The validated substrate candidates ERBB2 and CD68 were among the top 10 proteins (Supplementary Fig. 15b). For the new HC substrates, we considered only disease and pathway links not previously linked to γ-secretase or its substrates ("new links"). Compared to a set comprising all new HC substrates and all known substrates ($n = 307 = 160 + 147$; Supplementary Fig. 15a), the new HC substrates constituted over 50% of the proteins, but their new links to pathways ($n = 157$), diseases ($n = 313$), and mutations within the TMD-JMD ($n = 39$) accounted for only 7, 9, and 31% of all links, respectively (Fig. 10a). Most of the 313 new disease links belonged to the DisGeNET class neoplasms ($n = 40$; EGFR, ERBB2, KITLG, AREG) but also to immune diseases ($n = 29$; FAS), neurological diseases ($n = 13$; PNPLA6), and cardiovascular diseases ($n = 13$; CD163) (Fig. 10b, c). The largest number of disease/pathway links was found for the apoptosis-mediating FAS ($n = 52$) and for two cancer-related proteins, EGFR ($n = 43$) and ERBB2 ($n = 39$) (Fig. 10c).

Compared to the three other confidence-based substrate classes, HC substrates had a significantly higher ($P < 0.001$, two-sided Mann–Whitney U-test with Bonferroni correction) number of pathway and disease links (Supplementary Fig. 15c). They also generally showed significantly greater ($P < 0.001$) network-based measures including degree, stress centrality, and neighborhood connectivity, suggesting roles as network hubs, bottlenecks, or module members, respectively (Fig. 10d, Methods "Network analysis"). Moreover, HC substrates also exhibited a significantly higher ($P < 0.001$) sequence and CPP feature similarity and displayed subcellular locations closely matching those of known substrates (Supplementary Fig. 15d, e).

To assess the likelihood of in vivo interactions between γ-secretase and its substrates, we analyzed their co-expression across various tissues and cell types. Compared to the other confidence-based substrate classes, the co-expression of γ-secretase with HC substrates was significantly higher ($P < 0.05–0.001$, tested as before) at both tissue and single-cell RNA levels, as shown for selected HC

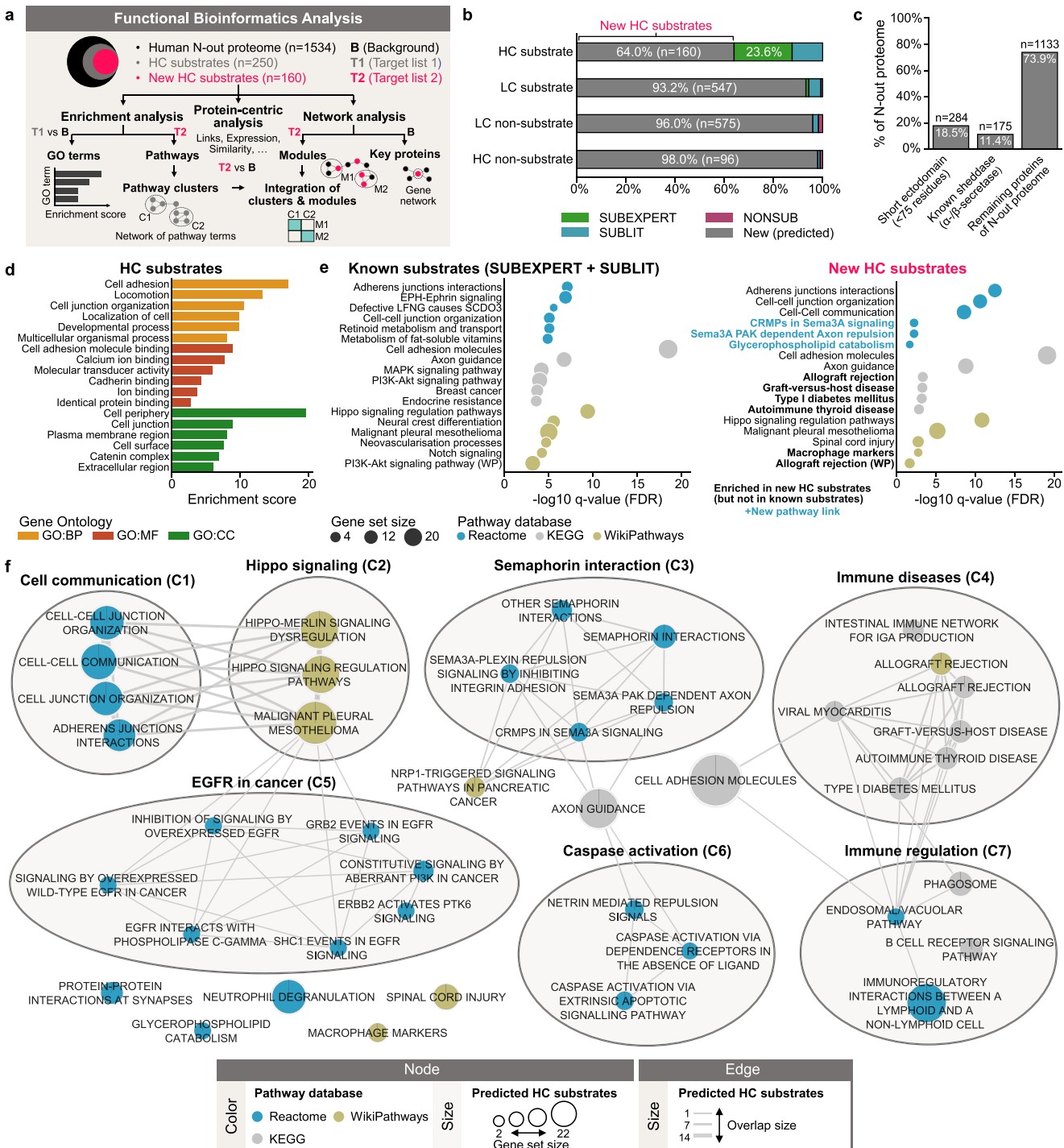

**Fig. 9 | Functional bioinformatics analysis of γ-secretase substrates by enrichment analysis and pathway clustering. a** Workflow of functional bioinformatics analysis for human single-span N-out membrane proteins (*n* = 1534, human N-out proteome). Separate enrichment analyses were performed for all 250 high-confidence (HC) substrates (gray) and the 160 new HC substrates (red), for which an additional network analysis was conducted. **b** Stacked bar chart showing the relative distribution within the human N-out proteome for proteins from the "New (predicted)" (gray), SUBEXPERT (green), SUBLIT (light blue), and NONSUB (purple) datasets across the four confidence-based substrate classes (see Methods "Confidence-based substrate classes"). The "New (predicted)" dataset comprises all proteins of the human N-out proteome with unknown substrate status. **c** Bar chart showing the number of proteins with a naturally short ectodomain or known sheddase in the human N-out proteome. **d** Gene ontology (GO) enrichment analysis results for all HC substrates compared to the human N-out proteome. Top

6 semantic clusters (see Supplementary Methods "Enrichment analysis") are shown for each GO domain: biological process (BP, orange), molecular function (MF, red), cellular component (CC, green). An enrichment score was computed for each semantic cluster as the mean −log10 *P* value of its constituent GO terms. **e** Pathway enrichment analysis results for known substrates (left) and new HC substrates (right) compared to the default g:Profiler background. The Benjamini-Hochberg adjusted −log10 *P* values are shown for the top 5–6 pathway terms from Reactome (light blue), KEGG (gray), and WikiPathways (gold). New pathway links (i.e., terms not previously linked to γ-secretase or its substrates) are highlighted in light blue. **f** Map displaying 7 clusters (C1–C7) of pathway terms linked by shared genes. Nodes represent pathway terms, sized by the number of associated new HC substrates and color-coded as in (**e**). Edges indicate the size of gene set overlaps. See Supplementary Figs. 13–15 for results of further functional bioinformatics analysis. Source data are provided as a Source Data file.

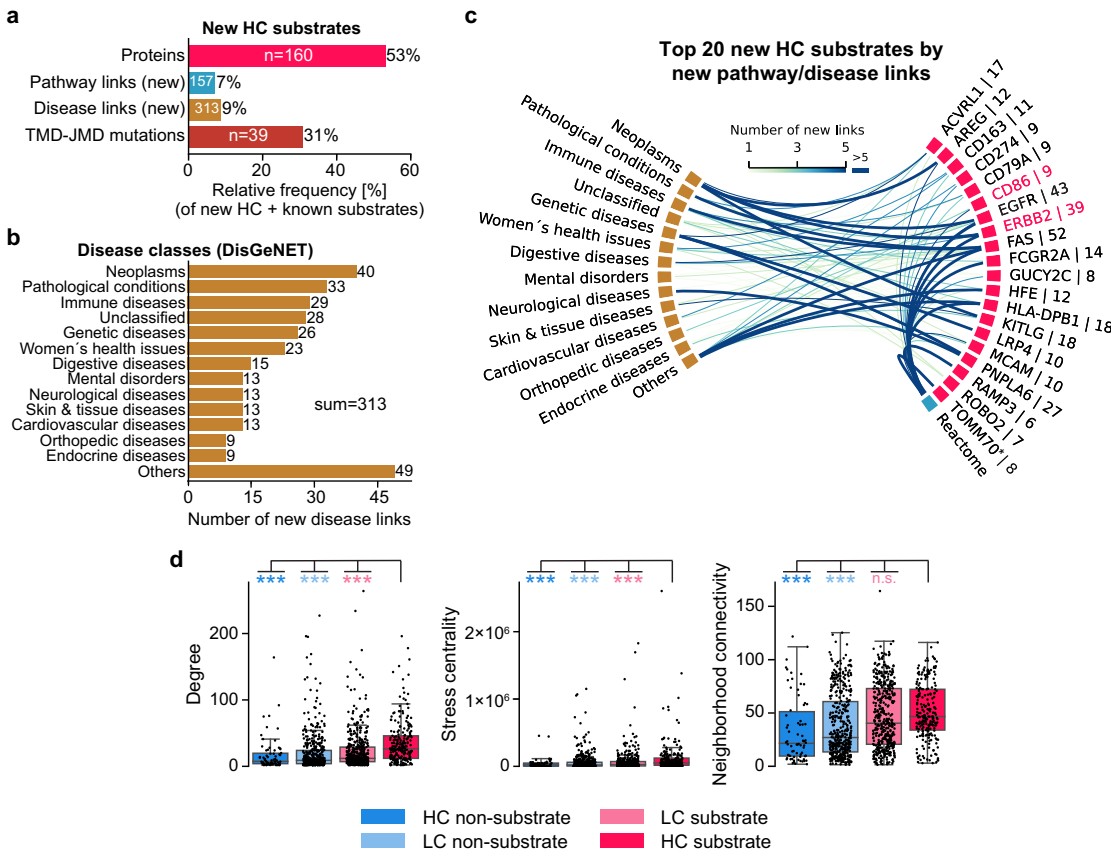

**Fig. 10 | Analysis of pathway, disease, and mutation links as well as network-based measures. a** Bar chart showing the percentage of three relevance factors for the set of new high-confidence (HC, $n = 160$ proteins) substrates compared to a set comprising all known substrates and new HC substrates. Total numbers are indicated within the respective bars. Only pathway and disease links were considered that were not linked before to γ-secretase or its known substrates. **b** Bar chart showing the number of disease links from (**a**), as classified by disease classes from DisGeNET. **c** Chord diagram illustrating the number of links for new pathways and disease terms from (**b**) for the 20 new HC substrates with the highest number of new links (Supplementary Data 17). Substrates are sorted alphabetically, and

validated ones are highlighted in red. Note that mitochondrial proteins, such as TOMM70 (highlighted in (**c**) by an asterisk), although unlikely γ-secretase substrates, were not excluded to keep our analysis unbiased. **d** Box plots for network metrics comparing the four confidence-based substrate classes (see Methods "Confidence-based substrate classes" for sample sizes). Differences between HC substrates and the other classes were tested using a two-sided Mann–Whitney $U$-test with separate Bonferroni correction. Significance levels are indicated by color-coded asterisks (*$P < 0.05$, **$P < 0.01$, ***$P < 0.001$). Source data are provided as a Source Data file.

substrates (Supplementary Fig. 15f–h, Methods "Co-expression analysis"). This correlation, exemplified by the brain-associated HC substrate PTPRD (Pearson's $r = 0.80$, $P < 0.001$; Supplementary Fig. 15g), suggests the co-evolution of γ-secretase and HC substrates. Taken together, although the functional repertoire of γ-secretase substrates has already been thoroughly characterized[8], our computational analysis reveals HC substrate candidates with new disease/pathway links and demonstrates roles of HC substrates as biological network hubs, bottlenecks, or module members (Supplementary Data 17).

## Discussion

To solve the problem of how γ-secretase recognizes its substrates, we developed CPP, a sequence-based bioinformatics method to identify discriminative physicochemical features. In addition, we developed dPULearn to address the issue of our imbalanced datasets. In combination, these algorithms allowed for the prediction of the γ-secretase substrate scope, including at least 160 substrate candidates. CPP and dPULearn reached a performance of 90% balanced accuracy in discriminating substrates from non-substrates, thereby outperforming state-of-the-art methods (deep learning-based protein embeddings combined with SMOTE, 57% balanced accuracy). While dPULearn shows 100% reproducibility, the advantages of CPP are that it is automated, alignment-free, and explainable/interpretable—i.e., CPP

reveals features constituting a physicochemical signature at single-residue resolution, going beyond consensus motifs.

CPP identified over 100 features in the TMDs of γ-secretase substrates and their flanking JMD regions. Remarkably, the substrate CPP feature map showed that amino acids with a high propensity to form α-helical and β-strand conformation are prominent in the region around the initial cleavage site. This observation is consistent with the α-helical conformation of the cleavage region of unbound substrates, which was shown to be extended in the γ-secretase enzyme–substrate complex for APP and NOTCH1[54,55]. In addition, residues with increased side-chain length, accessible surface area, and disorder are highly abundant in the TMD-C anchor—typically lysine and arginine. These amino acids are known to serve as membrane-anchoring residues[56], but our results show that they are more prominent in γ-secretase substrates. Interestingly, small residues are frequently found within the TMD-N, consistent with the requirement of flexibility in this region for certain substrates[11,13].

Substrate-defining features are particularly prevalent in HC substrates and decrease along the spectrum of substrate prediction scores, ultimately becoming absent in HC non-substrates. By combining CPP with SHAP, we demonstrated how the impact of these features varies across the substrate spectrum, enabling us to explain their substrate prediction scores at single-residue resolution for

**Table 1 | Key methods introduced in this work**

| Method | Description | Documentation link |
|---|---|---|
| CPP | Interpretable feature engineering algorithm | https://aaanalysis.readthedocs.io/en/latest/generated/aaanalysis.CPP.html |
| dPULearn | Deterministic positive-unlabeled learning method | https://aaanalysis.readthedocs.io/en/latest/generated/aaanalysis.dPULearn.html
https://aaanalysis.readthedocs.io/en/latest/generated/aaanalysis.dPULearnPlot.pca.html |
| CPP plotting functions | CPP ranking plot
CPP profile
CPP feature map | https://aaanalysis.readthedocs.io/en/latest/generated/aaanalysis.CPPPlot.ranking.html
https://aaanalysis.readthedocs.io/en/latest/generated/aaanalysis.CPPPlot.profile.html
https://aaanalysis.readthedocs.io/en/latest/generated/aaanalysis.CPPPlot.feature_map.html |
| Fuzzy labeling | Labeling technique for samples with an unknown class. | https://aaanalysis.readthedocs.io/en/latest/generated/aaanalysis.ShapModel.fit.html#aaanalysis.ShapModel.fit |
| CPP-SHAP plotting functions | CPP-SHAP ranking plot
CPP-SHAP profile
CPP/-SHAP heatmap | https://aaanalysis.readthedocs.io/en/latest/generated/aaanalysis.CPPPlot.ranking.html
https://aaanalysis.readthedocs.io/en/latest/generated/aaanalysis.CPPPlot.profile.html
https://aaanalysis.readthedocs.io/en/latest/generated/aaanalysis.CPPPlot.heatmap.html |

individual proteins. Based on our 80% cut-off, our data suggest that 16.3% ($n = 250$) of the entire human N-out proteome ($n = 1534$) are HC substrates of γ-secretase, of which 160 were not known previously. Whether these proteins are bona fide γ-secretase substrates is determined by additional factors such as subcellular localization or shedding prior to γ-secretase cleavage.

Our predictions are supported by the experimental validation of previously unknown substrates ($n = 11$) and non-substrates ($n = 4$) using both cell-based and cell-free biochemical assays, with an 88% success rate. All tested candidates with a score of >70% were experimentally confirmed as substrates, and all candidates scoring <30% as non-substrates. Among the validated substrate candidates are the cancer-related ERBB2 as well as the immune system-related CD2, CD68, and CD86. In addition, results of a large-scale proteomics screen of γ-secretase substrates at the endogenous level[49] align with our predictions (88% success rate). Finally, our functional bioinformatics analysis suggests that γ-secretase might be associated with a broader disease spectrum than known so far.

We anticipate that our approach offers a blueprint for identifying substrates of many other promiscuous proteases for which no clear consensus motifs exist. The suite of methods introduced here has the potential to advance the exploration of other molecular interactions, including antibody or receptor recognition.

## Methods

Further details for each computational method section are provided in the Supplementary Methods under the corresponding sections. Where applicable, additional Supplementary Methods sections are referenced at the end of the respective Method section.

### Data preparation

**Sequence parts of transmembrane proteins.** We focused on single-span transmembrane proteins (TPs) with an extracellular N-terminal ectodomain (i.e., an N-out topology). These proteins are characterized by an α-helical transmembrane domain (TMD) and can be distinguished based on the presence (type I) or absence (type III) of an N-terminal cleavable signal sequence. The following sequence parts were considered (Supplementary Fig. 1a, b):

- **TMD**: TMD as annotated in the UniProt database[37] or by the transmembrane prediction algorithms TMHMM[38] and Phobius[39].
- **TMD-N/C**: N- and C-terminal halves of TMD.
- **TMD-E**: TMD expanded by four amino acid positions on both sides of the membrane.
- **JMD-N/C**: N-terminal and C-terminal juxtamembrane domains with ten amino acids.
- **JMD-N-TMD-N**: A combined sequence of JMD-N and TMD-N.
- **TMD-C-JMD-C**: A combined sequence of TMD-C and JMD-C.
- **TMD-JMD**: A combined sequence of JMD-N, TMD, and JMD-C.

We denote the first four residues of the JMD-C of single-span type I TPs as "TMD-C anchor" (Fig. 2b). This region is typically characterized by positively charged residues "anchoring" the TMD at the membrane-water interface by electrostatic interactions with negatively charged phosphate groups[56].

**Datasets.** A dataset comprising 4464 single-span TP sequences (2365 from human and 2099 from mouse) was obtained from the UniProtKB/Swiss-Prot database[37], with missing UniProt topology information being supplemented by Phobius predictions. Since γ-secretase substrates have an N-out topology and are mainly of type I[6], we kept all type I TPs (containing 126 substrates[6,57–59] and 12 non-substrates[9,60,61]) and included three known non-substrates and 21 known substrates[6] from other organisms and/or with type III orientation, yielding 2179 proteins. After removing sequences with JMDs shorter than 10 residues, 2090 proteins remained. The dataset was further reduced to 670 proteins using the CD-HIT algorithm (40%-similarity cut-off)[62] applied to the TMD-JMD sequence, ignoring clusters with substrates or non-substrates.

We obtained the following datasets given for the UniProt annotation (with the corresponding number for the TMHMM annotation in parenthesis):

- **SUBSTRATES**: Set of 147 ($n = 136$ for TMHMM) known γ-secretase substrates, including 144 substrates from the most recent review[6], and three further substrates from other studies[57–59].
- **SUBEXPERT**: Non-redundant subset of SUBSTRATES with 68 ($n = 63$) expert-curated γ-secretase substrates selected if convincing evidence for cleavage was reported in ref. 6 and the corresponding primary literature.
- **SUBLIT**: Subset of SUBSTRATES with 79 ($n = 73$) literature-based γ-secretase substrates that were not selected for SUBEXPERT.
- **NONSUB**: Non-redundant set of 15 ($n = 14$) experimentally verified non-substrates from refs. 9,60,61.
- **OTHERS**: Non-redundant set of 670 ($n = 631$) single-span type I TPs with unknown substrate status.
- **NONSUBPRED**: Set of 53 ($n = 49$) predicted non-substrates identified from OTHERS (see "Computational non-substrate identification by dPULearn").

These datasets were generated by TMD annotations from UniProt, TMHMM, and Phobius predictions (Supplementary Fig. 1f and Supplementary Data 1, 2). The latter two mostly yielded slightly smaller datasets because of too short JMDs or missing TMD predictions. Unless stated otherwise, the TMHMM annotation was used because it allowed for the best prediction results (see "Aggregation of prediction results"). Since most γ-secretase substrates require shedding (i.e., removal of their N-terminal ectodomain) prior to γ-secretase cleavage, we compiled two lists of substrates for the two main sheddase families:

ADAM sheddases (α-secretases) as well as BACE1 and BACE2 (β-secretases) (Supplementary Data 3, 4).

One further dataset was collated for a functional bioinformatics analysis:

- **Human N-out proteome**: Set of 1534 human type I and type III single-span TPs with a JMD length of at least 10 amino acids each (Supplementary Data 17).

For the training of machine learning models, we assembled the following two datasets (given for the TMHMM annotation) of substrates and non-substrates used as positive and negative samples, respectively:

- **Dataset 1**: 63 substrates from SUBEXPERT as well as 63 non-substrates from NONSUB and NONSUBPRED.
- **Dataset 2**: 136 substrates from SUBEXPERT and SUBLIT as well as 63 non-substrates from NONSUB and NONSUBPRED.

**Sequence logos.** We used the WebLogo server[63] to create sequence motifs for 23 γ-secretase substrates with known cleavage sites, as well as 63 substrates from SUBEXPERT and 14 non-substrates from NONSUB (both based on TMHMM annotation).

## Feature engineering via Comparative Physicochemical Profiling (CPP)

**Idea of the CPP algorithm.** Comparative Physicochemical Profiling (CPP) is a sequence-based feature engineering algorithm to identify the most distinctive features between two sets of protein sequences. It amalgamates sequence segmentation techniques[64] with dis-/continuous motif identification[65] and *n*-gram methods[66]. A "CPP feature" is a part-split-scale combination (Supplementary Fig. 1a). CPP first splits a sequence part into smaller segments or patterns, and then assigns scale values to each residue to compute their mean values. These mean values are used to compare the two protein datasets.

By generating all possible part-split-scale combinations, CPP creates over 100,000 features and filters them statistically down to a user-defined number (default 100) of non-redundant features (Fig. 2a). CPP features are highly interpretable, leading to expressive machine learning models[30,31,67]. See Supplementary Methods ("Combining parts, splits, and scales") for further details.

**Splitting of sequence parts.** Sequence parts, such as the TMD, can be split into segments, patterns, or periodic patterns (Supplementary Fig. 1b, c). Segments are continuous subsequences of a sequence part, split into 1 to 15 equally sized segments. Patterns are discontinuous subsequences of a sequence part consisting of 2, 3, or 4 residues separated by 3 or 4 positions. Periodic patterns are discontinuous subsequences of a sequence part consisting of every third, fourth, or alternating third and fourth residue within a whole part. Both types of patterns represent the periodicity of an α-helix and potential interaction interfaces[33]. CPP generates in total 330 splits (Supplementary Fig. 1a, d; see part-split examples in Supplementary Data 5). See Supplementary Algorithm 1 for further details.

**Classification of scales.** A set of 652 amino acid physicochemical property scales[34,68,69] was assembled, reflecting crucial sequence-to-structure relationships[33,70–72]. We removed completely redundant scales and scales containing missing values, resulting in 586 scales. Each scale was min-max normalized to the [0,1] range. The classification of the property scales was retrieved from AAontology[35], classifying these 586 amino acid scales into 8 categories (e.g., conformation or energy) and 67 subcategories (e.g., coil or charge).

**Selection of scales.** Five sets of property scales (Set 1–5) were created in two steps. Set 1 contained all 586 scales, while sets 2 to 5 were subsets of Set 1. We first selected scale sets based on different scale classification criteria. Then, redundancy-reduced subsets of 2–5 were obtained using the AAclust framework[36] with agglomerative clustering (complete linkage). As AAclust selects one representative scale per cluster, we optimized the number of clusters such that each selected scale subcategory was contained at least once in the scale set (Supplementary Algorithm 2). Set 5, comprising 133 scales across 42 subcategories and 6 categories (Supplementary Fig. 1e and Supplementary Data 6, 7), showed the best benchmarking performance (Supplementary Fig. 4g) and was therefore chosen for subsequent analysis steps.

**CPP algorithm.** Taking two protein sequence datasets—a test set and a reference set—and a scale set as input, the CPP algorithm (Supplementary Fig. 1g) involves four steps:

1. **Feature creation**: All possible features for given parts, splits, and scales are created.
2. **Pre-filtering**: CPP removes features with a standard deviation in the test dataset higher than the threshold *max_std_test* and selects the top *pct_pre_filter* features with the highest mean difference between the test and reference dataset.
3. **Ranking**: All remaining features are ranked in descending order of the absolute adjusted area under the curve (AUC), which compares the reference and test sets. This adjusted AUC ranges from −0.5 to 0.5, i.e., all values in the test set are smaller or higher than the values in the reference set, respectively.
4. **Feature filtering**: The remaining features are filtered for redundancy regarding scale categories, sequence positions (via *max_overlap*), and scale correlation (via *max_cor*) until the desired maximum number of features, specified by *n_filter* (default 100), remains.

Comparing SUBEXPERT (test set) against OTHERS (reference set), the CPP algorithm created 131,670 features for three parts (TMD, JMD-N-TMD-N, and TMD-C-JMD-C), 330 splits (120 segments, 182 patterns, and 28 periodic patterns), and 133 min-max normalized scales (Set 5). To efficiently pre-filter these features, *max_std_test* = 0.2 and *pct_pre_filter* = 0.05 were used, yielding 6,583 features. In the filtering step, a value of 0.5 (50%) was empirically chosen for *max_overlap* and *max_cor* to balance between too high redundancy (at values close to 1) and the removal of too many potentially complementary features (at values close to 0). Sets of 150 and 100 CPP features were tested for machine learning (Supplementary Data 8, 9), but subsequent steps will only be described for the 150 features, as they yielded the best performance (Supplementary Fig. 5d).

## Substrate prediction by machine learning

**Machine learning models.** To predict γ-secretase substrates, we used 10 different types of machine learning classification models, applying default settings except where specified. This selection included 4 tree-based models (e.g., random forest), 2 linear models, 1 kernel-based model (support vector machine), 1 neural network, and 2 ensemble models (Supplementary Data 10). For benchmarking, support vector machine and random forest were employed as validation models, as recommended for small datasets[41,73]. Leave-one-out cross-validation was used for validation unless stated otherwise. See Supplementary Methods ("Feature representation" and "Evaluation measures") for further details.

**Derivation of the optimal scale and part sets for CPP.** To select the best sets of scales and parts for CPP, we trained 20 = 5 × 4 support vector machine classification models. We combined 5 scale sets (Set 1–5, see "Selection of scales") and 4 sets of parts: (1) TMD; (2) TMD-JMD; (3) TMD, JMD-N-TMD-N, TMD-C-JMD-C; and (4) TMD, TMD-E, JMD-N-TMD-N, TMD-C-JMD-C. All models were trained on SUBEXPERT (*n* = 63) against NONSUB (*n* = 14), using balanced accuracy as a

performance measure. The combination of scale set 5 ($n = 133$) and part set 3 yielded the highest balanced accuracy (84%, Supplementary Fig. 4g) and was therefore used for further analysis steps.

**Computational non-substrate identification by dPULearn.** To balance the training dataset of 63 known substrates (SUBEXPERT) and 14 non-substrates (NONSUB), we utilized positive-unlabeled (PU) learning[24,28,29]. Since common PU learning approaches[28,40,74] lead to irreproducible results due to their non-deterministic nature (see Supplementary Methods "Benchmarking dPULearn"), we developed a deterministic PU learning algorithm called dPULearn. dPULearn (Supplementary Fig. 3) first compresses the feature space using principal component analysis (similar to ref. 75) and then iteratively identifies putative non-substrates based on the $m$ principal components (PCs) with the highest explained variance. For each PC, dPULearn computes the average PC value (mean $PC_i$) for positive labeled proteins (SUBEXPERT) and selects the unlabeled proteins (OTHERS) with the greatest distance to mean $PC_i$ as additional non-substrates, where the number of selected proteins depends on the explained variance of the respective PC. Using dPULearn, we identified 49 additional non-substrates (NONSUBPRED), extending NONSUB ($n = 14$). Different TMD annotations showed a moderate overlap in the NONSUBPRED sets, with 53 and 51 additional non-substrates identified for the UniProt and Phobius annotations, respectively (Supplementary Data 11).

**Benchmarking of CPP and dPULearn.** We first compared dPULearn against a popular PU learning framework by ref. 40 regarding prediction performance and reproducibility (Supplementary Fig. 4a–f). Next, we evaluated CPP without and with NONSUBPRED identified by dPULearn, observing an improved prediction accuracy from 84 to 92% when NONSUBPRED was included (Supplementary Fig. 4g, h). We then benchmarked the performance of CPP and dPULearn against state-of-the-art protein embeddings (ProtT5)[25,26] utilized by transfer learning[27,30] (Fig. 3). The Synthetic Minority Over-sampling Technique (SMOTE)[42,43] was tested as an alternative to dPULearn for data expansion of the non-substrates. See information for each benchmarking step in Supplementary Methods ("Benchmarking dPULearn", "Benchmarking CPP without and with NONSUBPRED", and "Benchmarking CPP and dPULearn against deep learning-based embeddings") as well as benchmarking results in Supplementary Data 12.

**Training datasets.** Our machine learning pipeline (Supplementary Fig. 5a) was performed separately for each dataset-annotation combination. Two training datasets (dataset 1 and dataset 2) were collated for each TMD annotation (UniProt, TMHMM, Phobius). For the TMHMM annotation (Supplementary Fig. 5b), dataset 1 contained 63 substrates (SUBEXPERT) and 63 non-substrates (14 from NONSUB and 49 from NONSUBPRED), while dataset 2 comprised 136 substrates (63 from SUBEXPERT and 73 from SUBLIT) and the same 63 non-substrates as in dataset 1.

**Learning strategy.** We performed 25 independent training rounds (Supplementary Fig. 5a) to obtain a Monte Carlo estimate of prediction scores[76]. In each training round, a dataset was randomly split into a training set (80%) and a test set (20%), both containing a balanced proportion of substrates and non-substrates. As recommended for small datasets[41], we used a nested cross-validation approach, where the training set was used for feature selection and hyperparameter optimization by a 5-fold cross-validation. The test set was then used for an independent evaluation of the optimized models at the end of each round. See Supplementary Methods ("Feature selection" and "Model optimization and evaluation") for further details.

**Aggregation of prediction results.** We aggregated the best-performing training approaches for each dataset-annotation combination, selecting $6 = 2 \times 3$ approaches—corresponding to 2 datasets and 3 TMD annotations (UniProt, TMHMM, Phobius). To this end, each dataset-annotation combination was optimized for 6 feature pre-selection setups (Supplementary Fig. 5c). For each combination, we chose the approach with the highest average accuracy (Supplementary Fig. 5d, f), each comprising 250 trained models, from which an average prediction score was derived. Aggregating these scores across the 6 selected approaches ($1500 = 6 \times 250$ trained models, Supplementary Fig. 5e) yielded the final "substrate prediction score", with the standard deviation computed over the prediction scores of the 6 best approaches (Fig. 4d–f and Supplementary Data 13). Training on dataset 1 with TMHMM annotation resulted in the highest accuracy (96%, Supplementary Fig. 5d) and was thus used in subsequent steps unless otherwise stated.

**Confidence-based substrate classes.** Based on the substrate prediction score [0–100%], we classified single-span TPs into the following four classes, distinguished by varying prediction confidence:

- **HC substrate:** High-confidence substrate, prediction score ≥80%.
- **LC substrate:** Low-confidence substrate, prediction score ≥50 and <80%.
- **LC non-substrate:** Low-confidence non-substrate, prediction score <50 and >20%.
- **HC non-substrate:** High-confidence non-substrate, prediction score ≤20%.

Applying these confidence-based classes to the human N-out proteome ($n = 1534$) yielded 250 HC substrates, 587 LC substrates, 599 LC non-substrates, and 98 HC non-substrates (Fig. 4b).

## Experimental validation of predicted substrates and non-substrates

**Selection of substrate and non-substrate candidates.** Substrate and non-substrate candidates (human or murine) for experimental validation were selected primarily based on their substrate prediction score (Supplementary Data 13, 14). To gain further insight into cellular functions of γ-secretase, candidates were also selected using information derived from our functional bioinformatics analysis of the human N-out proteome (see "Computation of relevance score"). We favored candidates known to be cleaved by sheddases such as BACE1 or ADAM proteases (Supplementary Data 3, 4) to facilitate an unambiguous validation. Alternatively, candidates with an ectodomain shorter than 30 amino acids were also selected, assuming that they do not to require shedding prior to γ-secretase cleavage. For these proteins, the accumulation of their full-length (FL) form was used as a readout of substrate status.

**Cell-based cleavage assays.** The cleavage of candidates by γ-secretase was tested using transient overexpression of C-terminally 10×His-tagged proteins in HEK293 cells stably expressing APP carrying the Swedish mutation (HEK293/sw) and corresponding PS1/PS2 DKO cells[45] (Supplementary Fig. 7a). cDNA ORF clones of the candidates in pCMV3-C-His mammalian expression vector were purchased from Genomics online or Sino Biological. After 48 h of transient transfection using Lipofectamine 2000 (Invitrogen), levels of FL protein and its C-terminal fragment (CTF) (for type I TPs), or FL protein alone (for type III TPs), were analyzed by immunoblotting of cell lysates separated on Novex 10–20% Tris-Tricine gels (Invitrogen) using rabbit monoclonal anti-His tag antibody RM146 (biotin conjugate, NSJ Bioreagents, Catalog No. R20255BTN-50UG). In addition, cleavage of candidates was assessed in HEK293/sw cells 24 h following transient transfection by inhibition of γ-secretase overnight using 2 μM L-685,458[77] (Merck). To identify the FL protein of our candidates in the immunoblot analysis, the expected molecular weight (MW) of the (non-glycosylated) FL protein was calculated

from its amino acid sequence, not including the N-terminal signal sequence. The expected MW of the CTF was calculated from the sequence of the intracellular domain and the TMD (UniProt annotation) plus 15 adjacent extracellular amino acids similar to the ectodomain length of canonical γ-secretase substrates such as APP[78] or NOTCH1[79] (Supplementary Data 14). Comparable sample loading was confirmed by reprobing the immunoblots with mouse monoclonal anti-β-Actin antibody (Sigma, Product No. A5316, Batch number 123M4876).

**Cell-free cleavage assays.** TMD-based peptides of selected substrate and non-substrate proteins of γ-secretase were synthesized by Peptides Specialty Laboratories (Heidelberg, Germany). The peptides comprised the entire TMD combined with the first amino acid at the flanking N-terminal JMD and the first three amino acids of the flanking C-terminal JMD, of which the last amino acid was C-terminally tagged with biotin (Supplementary Fig. 8c). The peptide substrates were reconstituted in large unilamellar vesicles (LUV) composed of palmitoyl-oleoyl-phosphatidylcholine (POPC) at a 50:1 lipid/protein molar ratio by co-mixing an accurately weighed amount of 500–1000 μg peptide with the corresponding amount of POPC in 1 ml hexafluoroisopropanol (HFIP). After evaporation of HFIP with a gentle stream of nitrogen gas, the mixture was redissolved in 1 ml cyclohexane. Subsequent removal of cyclohexane by 2 h incubation in a SpeedVac concentrator resulted in a fluffy powder, which was suspended in ultrapure water (Sigma, Molecular Biology Reagent W4502) at a final peptide concentration of 200 μM. Following ten freeze-thaw cycles, LUVs were prepared by 21 extrusions through a 100-nm polycarbonate membrane and a LipofastTM extruder device (Armatis GmbH, Weinheim, Germany). The α-helical conformation of the reconstituted peptides was confirmed by circular dichroism by suspending the LUV-reconstituted peptides at 25 μM in water (Chirascan V100, Applied Photophysics, UK). Size and homogeneity of the LUVs were checked by dynamic light scattering (Zetasizer nano, Malvern Instruments), confirming the Z-average size of about 100 nm and a PDI <0.2. Finally, the LUV-reconstituted peptides were diluted with water, 1.5 M citrate (pH 6.4), and 32% (v/v) glycerol to a final concentration of 150 μM peptide in 30 mM citrate (pH 6.4), 3.5% (v/v) glycerol and stored in aliquots of 5 μl at −20 °C. For the cell-free cleavage assays (Supplementary Fig. 8a), aliquots of 20 μl containing POPC-reconstituted endogenous γ-secretase purified from HEK293 cells[80] were assayed for γ-secretase cleavage of LUV-reconstituted peptides at 2 μM (APP, APLP2, NOTCH1, NOTCH2, ERBB2, and SLC27A1) or 3 μM (ITGB1) final peptide concentration. Samples were incubated for 18 h at the indicated temperature with or without 2.5 μM DAPT[81] (Merck) with 300 rpm agitation in an Eppendorf ThermoMixerC (with ThermoTop) and subsequently analyzed by immunoblotting. For this, biotin-tagged substrate peptides and their cleavage products were separated by 16% Tricine SDS-PAGE[82] prepared with 40% acrylamide/bis-acrylamide 19:1, 5% crosslinker solution (Biorad). Immunoblots were blocked with biotin-free StartingBlock (ThermoFisher) and decorated with Immuno Pure Goat anti-biotin antibody (Pierce Biotechnology, Product No. 31852, Lot number EG769216).

# Explainable AI
## Explainable AI using SHAP
To enhance the interpretability of our machine learning models, we used the explainable AI framework SHapley Additive exPlanations (SHAP)[32,83]. Four key concepts of the SHAP framework are defined as follows:

- **Feature impact**: Positive or negative contribution of a feature, resulting in the model output for a sample (i.e., prediction score) to be higher or lower, respectively.
- **Base value**: Average model output of SHAP values over the entire training dataset.

- **SHAP output**: Sample-specific sum of the base value and all respective feature impacts, approximating the prediction score of a given sample.
- **Feature importance**: The absolute value of the feature impact, used for feature ranking.

**Combining CPP with SHAP.** To determine the impact of CPP features, we used the SHAP tree-based explainer[32,83] for computing SHAP values for the best features and the best tree-based models for each of the 25 training rounds (Supplementary Fig. 9a). Each tree-based model was re-trained on the complete training dataset (dataset 1 or dataset 2), and mean SHAP values were computed for each feature and sample across all models and rounds. To obtain the feature impact for a protein contained in the training dataset, we normalized its average SHAP values by dividing each by the sum of its absolute SHAP values. The feature importance was calculated by averaging the absolute SHAP values over all samples.

We then combined the CPP feature concept with SHAP values to reveal the residue-specific[84] feature impact for individual sequences, developing four visualizations: "CPP-SHAP ranking plot", "CPP-SHAP profile", "CPP heatmap", and "CPP-SHAP heatmap".

To obtain the feature impact for any protein not contained in the training dataset (hence unlabeled), we developed "fuzzy labeling". In this procedure, the unlabeled protein was included in the initial training dataset during model training, and its frequency of being labeled positive (as a substrate) or negative (as non-substrate) corresponded to its substrate prediction score (Supplementary Fig. 10a, b). See Supplementary Methods ("CPP-SHAP plots", "Fuzzy labeling", and "Clustering based on the feature impact") and Supplementary Data 15, 16 for further details.

## Functional bioinformatics analysis of γ-secretase substrates
**Enrichment analysis.** Enrichment analysis (Fig. 9a, b) for the human N-out proteome (n = 1534) was performed using the g:Profiler web server[85] with settings recommended in ref. 86. Derived GO term were semantically clustered using REVIGO[87] (similarity ≥0.5, default settings). The significant pathway terms (Reactome, KEGG, WikiPathways) were clustered and visualized using Cytoscape (version 3.9.1)[88] and EnrichmentMap[89] (edge similarity ≥0.5, node q value ≤0.05). Clusters were automatically named using the MCL clustering algorithm of the Cytoscape AutoAnnotate plugin and then manually refined for biological consistency. See the datasets and results of this computational analysis in Supplementary Data 17–19. See Supplementary Methods ("Dataset of human N-out proteome") for further details.

**Network analysis.** DOMINO web server[90] was used to identify protein modules for the new HC substrates (n = 160, Fig. 9b) based on the full STRING network[91]. We visualized the protein modules using Cytoscape[88] and integrated them with the clustered pathway terms. A whole network analysis was performed for the human N-out proteome on the STRING network obtained by the Cytoscape StringApp[92] (confidence ≥0.4, default; 0.8 was also tested). Using Cytoscape NetworkAnalyzer[93], we obtained the degree, neighborhood connectivity, and stress centrality of each protein in the network (Supplementary Data 17).

**Analysis of pathway, disease, and mutation links.** We downloaded Reactome pathway links from the g:Profiler web server, disease links from DisGeNET[94] (confidence score ≥0.1), and mutation links from the UniProt database[37] (Supplementary Data 17). We kept only mutations within the TMD-JMD reported in UniProt or the dbSNP database[95], of which several are disease-associated.

**Computation of relevance score.** For the new HC substrates, we obtained "new links" with pathways and diseases (Supplementary

Fig. 15a and Supplementary Data 20) not previously associated with γ-secretase or proteins in SUBEXPERT and SUBLIT, referred to as "known substrates". A "relevance score" was computed based on five "relevance factors": (a) existence of a "new pathway link"; (b) existence of a "new disease link"; (c) existence of mutations within the TMD-JMD sequence ("mutated TMD-JMD"); (d) whether the protein family to which the respective protein belonged to was not contained in protein families of the known substrates ("new protein family"); and (e) whether the TMD-JMD sequence did not exhibit more than 30% sequence identity to any substrate from SUBEXPERT ("dissimilar TMD-JMD"). Each relevance factor was assigned a value of 1 or 0 (true or false), and the relevance score was computed as their average (Fig. 4g and Supplementary Data 17).

**Comparison of CPP with a similarity-based approach.** Sequence similarity (whole sequence or TMD-JMD) was assessed using the BLAST algorithm[96]. The similarity between proteins from the human N-out proteome to the closest (i.e., most similar) substrate from SUBEXPERT was used as a similarity measure. Alternatively, Pearson correlation based on the top 100 CPP features was used (Supplementary Data 17).

**Co-expression analysis.** RNA expression data were obtained from the Human Protein Atlas database (version 21.1; Supplementary Data 21). The co-expression relationship between γ-secretase and proteins of the four substrate classes were assessed by Pearson correlation.

### Statistics
Differences between the HC substrates and the other substrate classes were tested by a two-sided Mann–Whitney $U$-test. $P$ values were adjusted by Bonferroni correction. See exact $P$ values and summary statistics for all performed tests in Supplementary Data 22. Analyses were conducted in Python v3.9 using key packages including pandas v2.2.1, SciPy v1.8.1, matplotlib v3.5.2, scikit-learn v1.1.1, and SHAP v0.44.0. See the Reporting Summary for a complete list.

### Reporting summary
Further information on research design is available in the Nature Portfolio Reporting Summary linked to this article.

## Data availability
The data generated in this study are provided in Supplementary Data 1–22 (overview in Supplementary Data 23). A subset of a previously published proteomics dataset[49] was used in this study and is included in our Source Data for transparency. Unless otherwise stated, all data supporting the results of this study can be found in the article, supplementary, and Source Data files. Source data are provided with this paper.

## Code availability
The methods introduced here (CPP and dPULearn) form the foundation of AAanalysis, a Python-based framework for interpretable, sequence-based protein prediction. AAanalysis is fully documented on Read the Docs [https://aaanalysis.readthedocs.io/en/latest/index.html] and freely available on GitHub [https://github.com/breimanntools/aaanalysis]. Tutorials can be found at [https://aaanalysis.readthedocs.io/en/latest/tutorials.html]. All analyses presented in this study were conducted using AAanalysis v1.0.0, which is installable via PyPi and archived on Zenodo [https://doi.org/10.5281/zenodo.15320204] for long-term access and reproducibility. Table 1 provides an overview of the primary algorithms and visualizations introduced in this work, along with a brief description and a link to their respective documentation.

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

## Acknowledgements

We are grateful to Michael Heinzinger and Burkhard Rost for their support with the application of ProtT5 protein embeddings and for providing critical manuscript feedback, and to Fabian Scheipl, Alexander Herzog, and Stefan Kramer for fruitful discussions on statistical ranking, explainable AI, and positive-unlabeled learning, respectively. We thank Martin Ortner, Regina Fluhrer, Matthias Voss, and Akio Fukumori for discussion of unpublished results, Christina Scharnagl for valuable discussions, and Christian Haass for support, stimulating discussions, and critically reading the manuscript. This work was funded by the Deutsche Forschungsgemeinschaft (DFG) through projects within the FOR2290 research network (ID 263531414; to S.F.L., D.L., D.F., and H.S.) and under Germany's Excellence Strategy within the framework of the Munich Cluster for Systems Neurology (EXC 2145 SyNergy – ID 390857198; to S.F.L.).

## Author contributions

S.B., F.K., D.F., and H.S. conceived and designed the study, discussed the results, and wrote and edited the manuscript. S.B. performed all bioinformatic analyses. G.B. carried out cell-based cleavage assays. C.A.-A. and F.K. performed cell-free cleavage assays. F.K. and H.S. supervised and analyzed cleavage assay data together with G.B. and C.A.-A. K.Y. and M.O. generated and provided PS1/PS2 double knockout cells. G.G. and S.F.L. identified previously unknown non-substrates. D.L. provided critical input on data analysis. S.A.M. supported the biological interpretation of the functional bioinformatics analysis. S.B. and F.K. prepared the figures. D.L., S.A.M., and S.F.L. commented on and contributed to manuscript editing. D.F. and H.S. supervised the overall study.

## Funding

## Competing interests

The authors declare no competing interests.
