## [Transparent Peer Review file · Nature Communications]

Charting γ -secretase substrates by explainable AI

Corresponding Author: Dr Harald Steiner

Version 0:

Reviewer comments:

Reviewer #1

(Remarks to the Author)

This paper introduces a sequence-based bioinformatics approach to identify substrates of γ -secretase, incorporating a positive-unlabeled learning strategy to enhance negative sample representation, particularly addressing the challenges posed by unbalanced datasets. The model demonstrates superior performance compared to protein language models, further supported by robust experimental validation of novel substrates. The methodology is clearly articulated, and the manuscript is well-written and accessible, making it a candidate for publication. However, several concerns must be addressed before the manuscript can be accepted, as detailed below.

Comments:

- (1) As for the machine learning workflow for γ -secretase substrate prediction, the predictions in this study are based on aggregated scores from 1,500 trained models. However, these models are trained on a small dataset consisting of only a few hundred substrate samples. Given the limited data, the computational cost of building such an ensemble might outweigh its benefits. There is also a significant risk that the ensemble could collectively overfit to the small dataset, resulting in good performance on the training data but poor generalization to unseen data.
- (2) In the section on feature engineering using CPP, the authors mention that "sequence parts—in this case, the TMD and its adjacent N- and C-terminal juxtamembrane domains (JMD-N and JMD-C) of single-span membrane proteins—can be split into either continuous segments or discontinuous patterns, reflecting helical periodicity (Extended Data Fig. 1b–d)." The rationale behind this splitting of sequence parts to reflect helical periodicity is unclear. Please clarify the reasoning for this operation, and provide further details on the "Comparative Physicochemical Profiling (CPP)" feature.
- (3) In the Methods section, the authors state, "To efficiently pre-filter these features, $\text{max_std_test}=0.2$ and $\text{pct_pre_filter}=0.05$ were used, yielding 6,583 features. In the filtering step, a value of 0.5 was empirically chosen for max_overlap and max_cor ." Please provide an explanation and justification for these threshold parameters.
- (4) In the section on Identification of additional non-substrates by dPULearn, the authors mention, "For each PC, proteins from OTHERS that are most distant to SUBEXPERT proteins are identified as additional non-substrates." How is the distance quantified, and how was the number of predicted non-substrates (NONSUBPRED) set to 49?
- (5) In the same section, the authors state, "CPP achieved 84% balanced accuracy with an optimized part set and 92% accuracy when NONSUBPRED was included to balance the datasets." Is the reported model accuracy calculated with or without the inclusion of NONSUBPRED in the dataset?
- (6) In the section on Identification of additional non-substrates by dPULearn, the manuscript states, "10 different machine learning classification model types—4 tree-based, 2 linear, 1 kernel-based, 1 neural network, and 2 ensemble model classes—were used in 25 training rounds, yielding $250=25 \times 10$ trained models." The authors should clarify whether each model contributed equally to the overall performance in substrate identification. Validation is necessary to illustrate the effectiveness of each model.
- (7) In the Section of Feature ranking using machine learning, the authors selected the 6 best approaches for different dataset-annotation combinations, resulting in $1,500=6 \times 250$ trained models. Have the authors investigated how the number of models in the ensemble affects performance? Is such a large ensemble essential for substrate identification?
- (8) The section on "Feature Ranking Using Machine Learning" could be better integrated with the subsequent "Physicochemical Signature of γ -Secretase Substrates" section or moved to the Methods section, as no direct results or findings are presented here.
- (9) In the "Physicochemical Signature of γ -Secretase Substrates" section, the authors state, "The 10 most important CPP features comprise a cumulative feature importance of 28%, but over 100 of the top 150 features account for 100% (Fig. 1i)."

Does this imply that the last 50 features contribute minimally to γ -secretase substrate identification, while the top 100 features are sufficient?

(10) In the "CPP and dPULearn Outperform State-of-the-Art Methods" section, the authors compare their methods with a scale-based and a deep learning-based feature engineering approach. While the deep learning-based features were derived from ProtTrans5, more details on the scale-based embedding are expected.

(11) The sentence, "To ensure comparability, average values were computed over the entire TMD-JMD sequences for amino acid scales and embeddings," should be reorganized for clarity.

(12) In the same section, the authors refer to "embeddings" in the context of feature engineering methods (scale-based, embeddings, CPP) and data expansion techniques (None, SMOTE, dPULearn). The term "embeddings" is unclear and requires further clarification.

(13) In the section CPP and dPULearn outperform state-of-the-art methods, the authors use support vector machine models with default settings to conduct a model performance comparison. A more fair comparison would involve using the same model architecture as in this study instead of the default support vector machine models.

(14) The analysis result shown in Figure 2b seems to be missing from the manuscript. Please provide more details about the feature number optimization.

(15) The fuzzy labeling strategy described in the "Explaining Prediction Results at the Amino Acid Sequence Level" section is confusing. It seems straightforward that the model tends to perform better on data it has seen during training than on unseen data, which is expected. Clarification is needed.

(16) The work related to discovering substrate subgroups based on feature impact in the "Explaining Prediction Results at the Amino Acid Sequence Level" section seems redundant, as it is straightforward that similar feature embeddings lead to similar model outputs (i.e., substrate probability).

(Remarks on code availability)

Reviewer #2

(Remarks to the Author)

This study employs AI to identify the substrate proteome of gamma-secretase. Although over 150 substrates have been identified to date, many additional substrates are believed to be at large due to the lack of sequence specificity: the enzyme appears to process the transmembrane domain of almost any N-out membrane protein with a short lumen/extracellular ectodomain (typically formed through a prior sheddase cleavage event). Only a handful of proteins have been identified as nonsubstrates, and the rules for what makes a substrate vs. a nonsubstrate (beyond a short N-terminal ectodomain) remain unknown.

By providing a training set of known substrates, the authors were able to develop a highly predictive algorithm for substrates and nonsubstrates through machine learning, validating the approach through biological assays. Importantly, the researcher team evaluated RNA levels at tissue and single-cell levels to show that newly identified substrates are indeed co-expressed with gamma-secretase. Although many new candidate substrates were identified, only a small fraction were tested biologically. Nevertheless, this study provides rankings/probabilities for all substrates, allowing other investigators to determine if their membrane protein of interest might be worthwhile to test as an actual substrate. The authors also conducted bioinformatics analyses to connect the new substrates to potential biological and pathological roles.

The algorithm ranks physicochemical and conformational features that are commonly found in substrates. Most notable is the ability of the region around the initial cleavage site (in the C-terminal region of the transmembrane domain) to readily assume an alpha helix or a beta-strand. This is consistent with the unbound transmembrane domain being helical (as seen by NMR for APP and Notch) but adjusting to a beta-strand in binding to the gamma-secretase active site (as seen in cryoEM structures of bound substrates).

Overall, this is an excellent study and represents a substantial advance in understanding the biology of gamma-secretase, an intramembrane protease complex central to metazoan biology and human disease.

(Remarks on code availability)

The code is available, but I am not able to assess the code, as my expertise lies in the biochemistry and biology of gamma-secretase, not computer codes.

Version 1:

Reviewer comments:

Reviewer #2

(Remarks to the Author)

I had no concerns with the original submission of this manuscript. Nevertheless, the authors have revised the study to include validation of additional substrates, addressing my comment on the limited number of newly identified substrates that were validated through biological experimentation. The original manuscript was fully acceptable in my opinion, for reasons explained in my first review, and the study is even better now.

(Remarks on code availability)

I do not have expertise in coding and cannot comment on this aspect of the work.

Reviewer #3

(Remarks to the Author)
See PDF

(Remarks on code availability)

Reviewer #4

(Remarks to the Author)
I co-reviewed this manuscript with one of the reviewers who provided the listed reports. This is part of the Nature Communications initiative to facilitate training in peer review and to provide appropriate recognition for Early Career Researchers who co-review manuscripts.

(Remarks on code availability)

Version 2:

Reviewer comments:

Reviewer #3

(Remarks to the Author)
The authors seem to have suitably addressed all concerns raised.

I initially assumed that the proteins below the threshold in the dataset from Hou et al. (2023) were non-substrates. However, as the authors' response states, substrates below this threshold are merely undetermined, which makes their validation method relevant.

The second main concern regarding the dPULearn benchmark was also addressed. The authors have tested their classifier on true non-substrates, yielding results similar to those of other methods.

(Remarks on code availability)

Reviewer #4

(Remarks to the Author)
I co-reviewed this manuscript with one of the reviewers who provided the listed reports. This is part of the Nature Communications initiative to facilitate training in peer review and to provide appropriate recognition for Early Career Researchers who co-review manuscripts.

(Remarks on code availability)

Point to point response to reviewer's comments

Reviewer #1:

This paper introduces a sequence-based bioinformatics approach to identify substrates of γ -secretase, incorporating a positive-unlabeled learning strategy to enhance negative sample representation, particularly addressing the challenges posed by unbalanced datasets. The model demonstrates superior performance compared to protein language models, further supported by robust experimental validation of novel substrates. The methodology is clearly articulated, and the manuscript is well-written and accessible, making it a candidate for publication. However, several concerns must be addressed before the manuscript can be accepted, as detailed below.

We thank the reviewer for his positive evaluation of our work. We have addressed all comments as follows below. Revisions made in the manuscript are highlighted in the following as excerpts of the revised manuscript with changes made highlighted in red.

Comment 1: As for the machine learning workflow for γ -secretase substrate prediction, the predictions in this study are based on aggregated scores from 1,500 trained models. However, these models are trained on a small dataset consisting of only a few hundred substrate samples. Given the limited data, **the computational cost of building such an ensemble might outweigh its benefits.** There is also a **significant risk that the ensemble could collectively overfit to the small dataset, resulting in good performance on the training data but poor generalization to unseen data.**

Response: Given the relatively small size of the training dataset, the computational cost of training the ensemble of models was minimal (see **Appendix 1** for detailed hardware specifications and computation time). This ensemble approach was chosen to ensure reproducibility and robustness in the results (see **answer to comment 7** including **Appendix 4** for further results and discussion).

While we acknowledge the limited size of our dataset, which increases the risk of overfitting, we have already implemented measures to mitigate this risk. Despite the small number of predicted substrates (n=12) and non-substrates (n=5) that were experimentally tested and validated with a success rate of 88%, our model showed a strong generalization on these unseen data, aligning well with computational accuracy of 90% (**Fig. 2**). To further assess generalization on unseen data, we now additionally validated our approach using an independent proteomic screening of endogenously expressed γ -secretase substrates from the literature (Hou et al., 2023). Using a stricter identification criterion (see **Appendix 2**), we correctly predicted 42 out of 48 substrates, resulting also in an 88% success rate. These 48 proteins included 28 proteins for which it was not known whether they are substrates or non-substrates, *i.e.*, unseen data points. 24 of these 28 proteins were predicted to be substrates, yielding a similar success rate of 86%. This independent validation underscores the model's ability to generalize beyond the initial training data. To illustrate the correlation between our predictions and the proteomics results, we revised two sections of the main text and added a new panel figure (**Fig. 4e**), as shown in the following excerpts:

Excerpts from the revised manuscript (main text, pages 9–10, 14, and 19):

(I) Experimental validation

...In line with our **results**, recent studies identified and biochemically validated additional γ -secretase substrate candidates **for which we determined substrate prediction scores** of ~50% (CD300A, MILR1, and TNFRSF1B⁴⁹), above 70% (TNR12⁴⁷), and above 90% (PTPRD⁵⁰ and PTPRT⁵¹) (**Supplementary Table 13**). **To assess our substrate predictions for endogenously expressed γ -secretase substrates, we**

used a recent proteomics screen that identified 85 substrate candidates in human microglia-like cells by pharmacological γ -secretase inhibition⁴⁹. We observed a significant positive Spearman correlation ($\rho=0.44$, $P<0.001$) between our substrate prediction scores and their reported fold changes of CTF accumulation (Fig. 4e). Using a log₂ fold change above 1 as identification threshold, 42 out of 48 proteins were predicted as substrates, yielding a success rate of 88%. These 48 proteins included 28 proteins of unknown substrate status, of which 24 were predicted to be substrates, yielding a similar success rate of 86%. These success rates are consistent with our experimental results (88%), both aligning with our computational accuracy of 90% (Fig. 2a).

(II) Discussion

... Among the newly identified and validated substrates are the cancer-related ERBB2 as well as the immune system-related CD2, CD68, and CD86. In addition, results of a large-scale proteomics screen of γ -secretase substrates at the endogenous level⁴⁹ align with our predictions (88% success rate). Finally, our functional bioinformatics analysis suggests that γ -secretase might be associated with a broader disease spectrum than known so far.

(III) Figure 4e

... e, Scatterplot showing the correlation between substrate prediction scores and recently reported log₂ fold change (FC) of CTF accumulation in the presence of semagacestat γ -secretase inhibitor against control, obtained for endogenously expressed proteins in human microglia-like cells⁴⁹. The maximum FC was used for proteins with CTF detection only during inhibition. Regression estimate (solid black line) with 95% confidence interval and the chosen substrate identification FC threshold (dashed grey line) are indicated.

Comment 2: In the section on feature engineering using CPP, the authors mention that "sequence parts—in this case, the TMD and its adjacent N- and C-terminal juxtamembrane domains (JMD-N and JMD-C) of single-span membrane proteins—can be split into either continuous segments or discontinuous patterns, reflecting helical periodicity (Extended Data Fig. 1b–d)." **The rationale behind this splitting of sequence parts to reflect helical periodicity is unclear.** Please clarify the reasoning for this operation and provide further details on the "Comparative Physicochemical Profiling (CPP)" feature.

Response: The rationale for splitting sequence parts into discontinuous patterns is to capture general helical periodicity, where residues spaced 3/4 amino acids apart typically align on the same side of the helix. These residues form key interaction interfaces and are potentially crucial for substrate recognition since all γ -secretase substrates are single-span membrane proteins with a helical transmembrane domain. Our CPP approach uses this biologically inspired heuristic to balance capturing meaningful structural patterns while avoiding a combinatorial explosion. Limiting distances to 3 and 4 residues yields 182 patterns (see **Extended Data Fig. 1d** and **Supplementary Table 5**), whereas including distances of 1–4 residues would create over 5,000 patterns. Distances of 3 to 4 residues can represent helical face interactions, short-range interactions in unstructured regions, or short-range interactions that allow transitions between extended and helical conformations.

CPP features are defined by part-split-scale combination, as explained in detail in the “Splitting of sequence parts” section of the *Supplementary Information*. Examples of the positions of the residues resulting from part-split combinations are generally highlighted in our CPP feature ranking plots (in grey), such as in **Figure 5 a–c** and in **Extended Data Figure 6c**.

Since CPP features are introduced in detail in the *Supplementary Information* and **Extended Data Fig. 1**, we added a cross-reference to this section in the main text to enhance clarity. Additionally, we included the following revisions to the main text and Methods section to clarify the concept of periodic patterns reflecting helical periodicity.

Excerpts from the revised manuscript (main text, pages 3–4, 26–27):

(I) Feature engineering using CPP

...Sequence parts—in our case, the TMD and its adjacent N- and C-terminal juxtamembrane domains (JMD-N and JMD-C, respectively) of single-span membrane proteins—can be split into either continuous segments or discontinuous patterns (**Extended Data Fig. 1b–d**, *Supplementary Information*). These patterns reflect helical periodicity, where residues spaced 3 or 4 positions apart align on the same side of an α -helix³³.

(II) Methods: Splitting of sequence parts

Sequence parts, such as the TMD, can be split into segments, patterns, or periodic patterns (**Extended Data Fig. 1b, c**). Segments are continuous sub-sequences of a sequence part, split into 1 to 15 equally sized segments. Patterns are discontinuous sub-sequences of a sequence part consisting of 2, 3, or 4 residues separated by 3 or 4 positions. Periodic patterns are discontinuous sub-sequences of a sequence part consisting of every 3rd, 4th, or alternating 3rd and 4th residue within a whole part. Both types of patterns represent the periodicity of an α -helix and potential interaction interfaces³³. CPP generates in total 330 splits (**Extended Data Fig. 1d**; see part-split examples in **Supplementary Table 5**).

Comment 3: In the Methods section, the authors state, “To efficiently pre-filter these features, `max_std_test=0.2` and `pct_pre_filter=0.05` were used, yielding 6,583 features. In the filtering step, a value of 0.5 was empirically chosen for `max_overlap` and `max_cor`.” Please **provide an explanation and justification for these threshold parameters**.

Response: We appreciate the reviewer’s request for clarification regarding the thresholds of the parameters used in the pre-filtering (`max_std_test`, `pct_pre_filter`) and filtering (`max_overlap`, `max_cor`) steps of the CPP algorithm. Below, we provide detailed explanations and justifications for these parameters:

- **max_std_test:** This parameter defines the maximum allowed standard deviation within the test dataset for feature pre-filtering. By setting *max_std_test*=0.2, we ensure that only features with low variability (maximum of 20%) within the test group are retained, reducing the risk of overfitting to noise by eliminating highly variable features that may lead to poor generalization.
- **pct_pre_filter:** This parameter specifies the percentage of features to retain after the pre-filtering step. We set *pct_pre_filter*=0.05 to retain the top 5% of features that exhibit the greatest discrimination between the test and reference groups (measured by absolute *AUC*), helping to focus the model on the most relevant features and improve the efficiency of the CPP algorithm. If the difference between the train and test datasets are weaker, this value may need to be increased.
- **max_overlap:** This parameter sets the maximum allowed positional overlap between features. With *max_overlap*=0.5, we limit redundancy by preventing the retention of features that overlap by more than 50% in their residue positions. A higher value would result in less strict filtering, increasing the number of retained features but also potentially introducing more redundant information.
- **max_cor:** This parameter defines the maximum allowed Pearson correlation between the property scales used for the features. Setting *max_cor*=0.5 ensures that features with high Pearson correlation (over 50%) regarding their used property scales are filtered out, maintaining only complementary and non-redundant features. Higher values would result in less strict filtering, leading to increased redundancy in the property scales used by CPP features.

The effect of different thresholds is demonstrated the CPP documentation and tutorial within the AAanalysis framework. These parameters, along with the effects of varying them, are detailed in the "CPP Algorithm" section of the *Supplementary Information*. We have also added a remark in the "CPP algorithm" *Method* section of the main text explaining the rationale for choosing *max_overlap* and *max_cor* at 0.5 (50%), as a balanced trade-off between avoiding high redundancy and preventing the removal of too many complementary features (see following excerpt). We additionally performed an analysis on the impact of varying CPP pre-filtering (*pct_pre_filter* and *max_std_test*) and filtering (*max_overlap* and *max_cor*) setting thresholds on the prediction performance, the processing time, and the number of obtained CPP features (see Appendix 3).

Excerpt from the revised manuscript (main text, page 27):

Methods: CPP algorithm

...To efficiently pre-filter these features, *max_std_test*=0.2 and *pct_pre_filter*=0.05 were used, yielding 6.583 features. In the filtering step, a value of 0.5 (50%) was empirically chosen for *max_overlap* and *max_cor* to balance between excessively high redundancy (at values close to 1) and the removal of too many potentially complementary features (at values close to 0).

Comment 4: In the section on Identification of additional non-substrates by dPULearn, the authors mention, "For each PC, proteins from OTHERS that are most distant to SUBEXPERT proteins are identified as additional non-substrates." **How is the distance quantified, and how was the number of predicted non-substrates (NONSUBPRED) set to 49?**

Response: The distance between proteins from OTHERS and SUBEXPERT proteins is quantified using the absolute differences in their principal component (PC) values. dPULearn applies principal

component analysis (PCA) to reduce the high-dimensional feature space onto principal components, each representing a dimension of variance. For each PC, the distance between a protein from OTHERS and the mean PC value of all SUBEXPERT proteins is calculated. Proteins from OTHERS with the greatest absolute distances from the SUBEXPERT mean for the respective PC (e.g., the mean for PC1) are selected as additional non-substrates.

The number of predicted non-substrates (NONSUBPRED) was set to 49 to balance the dataset. Given that there are 63 expert-curated substrates (SUBEXPERT) and 14 known non-substrates (NONSUB), we identified 49 additional non-substrates (NONSUBPRED) to equalize the total number of non-substrates and substrates at 63 each. This ensures a balanced dataset for training the machine learning model.

For further details on the algorithm and selection process, please refer to the "Computational non-substrate identification by dPU Learn" section of the *Supplementary Information*. To enhance the clarity of our manuscript, we have incorporated the answers to this comment into the main text, as shown in the following excerpt.

Excerpts from the revised manuscript (main text, page 4):

Identification of additional non substrates by dPU Learn

To overcome the challenge for robust machine learning posed by an unbalanced dataset comprising 63 substrates (SUBEXPERT) and 14 non-substrates (NONSUB), we developed a new deterministic PU learning algorithm (dPU Learn) for identifying additional negatives from unlabeled data based on CPP features (Extended Data Fig. 2a). dPU Learn uses principal component analysis to compress the entire feature space (*i.e.*, a $n \times m$ matrix, where n is the number of proteins and $m=150$ is the number of features) onto principal components (PCs). For each PC, proteins from OTHERS that are most distant to SUBEXPERT proteins are identified as additional non-substrates, **based on the absolute distance between their PC value and the mean PC value of SUBEXPERT proteins**. Using dPU Learn, we extended the set of 14 known non-substrates by 49 predicted non-substrates (NONSUBPRED), **balancing the dataset at 63 substrates and 63 non-substrates (Extended Data Fig. 2b–e, Methods)**.

Comment 5: In the same section, the authors state, “CPP achieved 84% balanced accuracy with an optimized part set and 92% accuracy when NONSUBPRED was included to balance the datasets.” **Is the reported model accuracy calculated with or without the inclusion of NONSUBPRED in the dataset?**

Response: We appreciate this important question and would like to clarify any confusion. The reported model performance of 84% balanced accuracy (**Extended Data Fig. 2l**) refers to the model trained and validated without the inclusion of NONSUBPRED, using only NONSUB and SUBEXPERT data. The 92% accuracy (**Extended Data Fig. 2m**) was achieved after NONSUBPRED was included to balance the dataset. NONSUBPRED was used for both training and evaluation, following the leave-one-out cross-validation approach, as detailed in the “Model optimization and evaluation” section of the *Supplementary Information*.

One of our key results, demonstrated in **Fig. 2a**, shows that including NONSUBPRED during training also improves performance when validated solely on NONSUB and SUBEXPERT. This result is described in the “CPP and dPU Learn outperform state-of-the-art methods” section of the main text. Since this outcome appears later in the manuscript (page 7), we chose not to cross-reference it to **Extended Data Fig. 2m** (cross-referenced in page 5) to maintain clarity and flow in the earlier sections.

Comment 6: In the section on Identification of additional non-substrates by dPULearn, the manuscript states, “10 different machine learning classification model types—4 tree-based, 2 linear, 1 kernel-based, 1 neural network, and 2 ensemble model classes—were used in 25 training rounds, yielding 250=25×10 trained models.” The authors should **clarify whether each model contributed equally to the overall performance in substrate identification**. Validation is necessary to illustrate the effectiveness of each model.

Response: We appreciate this important question. Each of the 10 machine learning model types contributed equally to the overall performance in substrate identification. The prediction scores from all models were directly averaged without weighting across the model types or the 25 training rounds, as indicated in **Extended Data Fig. 3a**. While there were slight variations in performance depending on the dataset and annotation, all models showed robust performance (**Extended Data Fig. 3f**). Among the models, the multi-layer perceptron (MLP) showed slightly weaker performance compared to the others. However, overall, the inclusion of diverse model types contributed to balanced and robust predictions. For identifying the feature importance, we utilized the 4 tree-based models and their permutation-based feature importance.

We hope this clarifies the reviewer's question and believe that no further clarification in the main text is necessary. The fact that we computed an unweighted average across all models is already described in the manuscript, as shown in the following excerpt.

Excerpt from the *original manuscript* (main text, page 7):

Substrate prediction using machine learning

The probability of a given protein being cleaved by γ -secretase (termed ‘substrate prediction score’) was computed as the average prediction score over the 6 best-performing dataset-annotation combinations, using a total of 1500=250×6 trained machine learning models (**Extended Data Fig. 3d–f, Methods**).

Comment 7: In the Section of Feature ranking using machine learning, the authors selected the 6 best approaches for different dataset-annotation combinations, resulting in 1,500=6×250 trained models. **Have the authors investigated how the number of models in the ensemble affects performance? Is such a large ensemble essential for substrate identification?**

Response: We continuously assessed how the number of models in the ensemble impacted prediction performance, substrate prediction score robustness, and feature importance throughout the project. Our primary goal was to ensure reproducibility, which we achieved through iterative improvements. As described in the “Learning strategy” section of the *Methods* part, we performed 25 independent training rounds, each involving dataset splits, feature selection, hyperparameter optimization, and model evaluation (**Extended Data Fig. 3a**). This provided a Monte Carlo estimate of prediction scores and the feature importance, ensuring robustness in our results.

Additionally, we incorporated different datasets and annotations to estimate uncertainty arising from biological variability. By training models on different TMD annotations (representing different biological assumptions), we gained a more comprehensive understanding of how these variations might affect substrate predictions, resulting in more reliable and generalizable outcomes.

While these optimizations were crucial to improving robustness, we chose not to include detailed analyses of how robustness scales with the number of training rounds and models in the main text to avoid excessive technical details. However, we now conducted three in-silico experiments to examine

the effects of the number of models on the (a) prediction performance, (b) reproducibility of feature importance, and (c) consistency of the substrate prediction score. These findings are detailed in **Appendix 4**.

Comment 8: The **section** on "Feature Ranking Using Machine Learning" **could be better integrated with the subsequent** "Physicochemical Signature of γ -Secretase Substrates" **section** or moved to the Methods section, as no direct results or findings are presented here

Response: We agree with the reviewer's suggestion to better integrate the "Feature ranking using machine learning" section with the "Physicochemical signature of γ -secretase substrates" section. To improve the flow, we added a transition that links the two sections by emphasizing the connection between feature ranking and the subsequent physicochemical analysis. Rather than moving this section to the *Methods*, we believe this revised structure better demonstrates the direct relevance of feature ranking to the interpretation of the physicochemical properties of substrates.

Excerpts from the revised manuscript (main text, page 6):

Feature ranking using machine learning

...For feature ranking, the feature importance was obtained directly from the 4 tree-based model types (100=25×4 trained models) and averaged (*Methods*, **Supplementary Table 8**). If not stated otherwise, results are described for dataset 1 with TMHMM annotation due to its best performance (**Extended Data Fig. 3d**). **These ranked features form the basis for distinguishing γ -secretase substrates from non-substrates.**

Physicochemical signature of γ -secretase substrates

To enable an insightful interpretation of the CPP features and their importance as obtained by machine learning models, we developed the 'CPP profile' and 'CPP feature map' visualizations (**Fig. 1f, g**).

Comment 9: In the "Physicochemical Signature of γ -Secretase Substrates" section, the authors state, "The 10 most important CPP features comprise a cumulative feature importance of 28%, but over 100 of the top 150 features account for 100% (Fig. 1i)." **Does this imply that the last 50 features contribute minimally to γ -secretase substrate identification, while the top 100 features are sufficient?**

Response: Yes, the top 100 (more precisely top 117) features are indeed sufficient for γ -secretase substrate identification, with the last 50 (more precisely last 33) features contributing minimally. We have revised the sentence for clarity to reflect this conclusion, as shown in the following excerpt.

Excerpt from the revised manuscript (main text, page 6):

Physicochemical signature of γ -secretase substrates

...**Of the 150 CPP features, the 10 most important ones constitute 28% of the cumulative feature importance, the top 50 account for 73%, and the top 117 for 100% (Fig. 1i), indicating that the last 33 features contribute minimally to γ -secretase substrate identification.** We refer to this set of CPP features and their importance as the common physicochemical signature of γ secretase substrates.

Comment 10: In the "CPP and dPULearn Outperform State-of-the-Art Methods" section, the authors compare their methods with a scale-based and a deep learning-based feature engineering approach. While the deep learning-based features were derived from ProtTrans5, **more details on the scale-based embedding are expected.**

Response: We appreciate the reviewer's request for more details on the scale-based feature engineering approach. Scale-based features were derived from manually curated physicochemical properties (scales), such as hydrophobicity, charge, and secondary structure propensity. These properties were averaged across the entire transmembrane (TMD) and juxtamembrane (JMD) sequences to create a single representative value for each scale. This differs from ProtTrans5 embeddings, which automatically capture sequence features without prior knowledge of specific scales. We have revised the manuscript to clarify these distinctions, incorporating the following related comments 11 and 12.

Comment 11: The sentence, "To ensure comparability, average values were computed over the entire TMD-JMD sequences for amino acid scales and embeddings," **should be reorganized for clarity.**

Response: We agree with the reviewer's suggestion and have reworded the sentence for clarity, as shown in the excerpt of the revised manuscript for comment 12.

Comment 12: In the same section, the authors refer to "embeddings" in the context of feature engineering methods (scale-based, embeddings, CPP) and data expansion techniques (None, SMOTE, dPULearn). **The term "embeddings" is unclear and requires further clarification.**

Response: "Embeddings" refer to numerical vector representations of protein sequences learned by deep learning models, specifically ProtTrans5 in this case. These embeddings capture structural and functional properties of the sequences automatically by leveraging patterns learned from large protein databases. In contrast, scale-based features are derived from predefined physicochemical properties. We have clarified this distinction in the revised manuscript, as shown in the following excerpt (also addressing comments 10 and 11).

Excerpt from the revised manuscript for comments 10–12 (main text, pages 6–7):

CPP and dPULearn outperform state-of-the-art methods

To **evaluate** how CPP and dPULearn perform against state-of-the-art protein prediction methods, we compared them with **both** a scale-based and a deep learning-based feature engineering approach, combined with the Synthetic Minority Over-sampling Technique (SMOTE)^{42,43} for handling imbalanced datasets. **Scale-based features were generated by averaging physicochemical properties (e.g., polarity, charge, volume) across the entire TMD-JMD sequence of a protein, creating for each scale a single representative value used as a feature. For the deep learning-based approach, we used the ProtTrans5 ('ProtT5') language model^{25,26}, which produced numerical vectors, known as embeddings, for each protein sequence. These embeddings represent scale-like residue properties learned by ProtT5 from large protein sequence datasets. To ensure comparability between the scale-based and embedding-based approaches, we also averaged the embedding values across the entire TMD-JMD sequence.**

Comment 13: In the section CPP and dPULearn outperform state-of-the-art methods, the authors use support vector machine models with default settings to conduct a model performance comparison. A **fairer comparison would involve using the same model architecture as in this study instead of the default support vector machine models.**

Response: We appreciate the reviewer’s suggestion and agree that using the same model architecture could offer an interesting alternative perspective. However, our primary goal in this comparison was to assess the general utility of different feature engineering approaches (scale-based, embeddings, and CPP) in a standardized machine learning setting. By using support vector machine (SVM) models with default settings, we ensured the same basic conditions where the performance differences could be attributed primarily to the feature engineering techniques rather than differences in model optimization or architecture. This allowed us to focus on how effectively each method translates raw data into meaningful features for prediction.

Additionally, SVM is a well-established baseline for binary classification, particularly suited to small datasets like ours, where overfitting is a concern. The default settings ensured that results weren’t overly influenced by hyperparameter tuning, which would vary across model architectures.

While we plan to explore the use of different models in future work, introducing them here would add unnecessary complexity and be out of the scope of this work. We believe that focusing on feature engineering with a standard SVM model provides a clear and interpretable comparison of the methods.

To reflect this discussion, we modified the current version of the manuscript, as shown in the following excerpt.

Excerpts from the revised manuscript (main text, page 7):

CPP and dPULearn outperform state-of-the-art methods

...For each combination of feature engineering methods (scale-based, embeddings, CPP) and data expansion techniques (None, SMOTE, dPULearn), support vector machine models with default settings were trained and consistently evaluated by leave-one-out-cross-validation on SUBEXPERT and NONSUB (*Supplementary Information*) **to compare the different approaches in a standardized baseline machine learning setting**. Support vector machine models employing scale-based feature engineering or embeddings showed only ~50% balanced accuracy without data expansion and ~60% with SMOTE (**Fig. 2a**).

Comment 14: The analysis result shown in **Figure 2b** seems to be missing from the manuscript. Please provide **more details about the feature number optimization**.

Response: We appreciate the reviewer’s attention to the details in Figure 2b. The feature number optimization process was conducted by evaluating different combinations of top CPP feature sets, with an increasing number of features used for model training and non-substrate identification. In total, 36 evaluations were performed (6x6 matrix), varying the number of features between 25 and 150 for both training and dPULearn (Fig. 2b). The optimization, validated by leave-one-out cross-validation, aimed to identify the best combination of top features that maximized balanced accuracy, achieving up to 90% with dPULearn when optimized. This process is detailed in the “Benchmarking CPP and dPULearn against deep learning-based embeddings” section of the *Supplementary Information* and is concisely reflected in the current version of the manuscript, as shown in the following excerpt.

Excerpt from the revised manuscript (main text, page 7):

CPP and dPU Learn outperform state-of-the-art methods

... Support vector machine models employing scale-based feature engineering or embeddings showed only ~50% balanced accuracy without data expansion and ~60% with SMOTE (Fig. 2a). Models employing embeddings and dPU Learn reached 65%. In contrast, models using CPP features achieved a balanced accuracy of 84% without data expansion and up to 90% with dPU Learn when optimized by testing different numbers of CPP features for model training and non-substrate identification with dPU Learn (Fig. 2b, Supplementary Information).

Comment 15: The fuzzy labeling strategy described in the "Explaining Prediction Results at the Amino Acid Sequence Level" section is confusing. It seems straightforward that the model tends to perform better on data it has seen during training than on unseen data, which is expected. **Clarification is needed.**

Response: We appreciate the reviewer's observation and agree that models generally perform better on data they have seen during training than on unseen data. However, the primary objective of fuzzy labeling is to provide reliable explanations for prediction scores of proteins not included in the initial training dataset. To achieve this, fuzzy labeling ensures that SHAP values can accurately approximate prediction scores for any protein with unknown prediction status (*i.e.*, those lacking ground-truth labels). Instead of using binary (positive or negative) labels, fuzzy labeling applies probabilistic labels inspired by fuzzy logic, capturing varying degrees of confidence to reflect prediction uncertainty.

In this approach, machine learning models are trained over multiple rounds, including proteins whose prediction scores need to be explained but which were not part of the initial training dataset. For these proteins, labels are assigned based on their prediction scores. For example, if a protein has a prediction score of 60% and fuzzy labeling is applied over 25 training rounds, it will be labeled as positive in 15 rounds and as negative in 10 rounds. This method can be applied to proteins with unknown prediction status—in our case, proteins with unknown substrate status (*i.e.*, not reported as substrate or non-substrate in the literature). Fuzzy labeling is especially useful when prediction scores are uncertain due to high variance, as seen for low-confidence substrates or low-confidence non-substrates (Fig. 3).

We showcased the application of fuzzy labeling for the Alzheimer's disease-relevant protein TREM2, which is a reported substrate included in SUBLIT but was predicted as a low-confidence non-substrate (36±20%). This uncertainty in the substrate prediction score of TREM2 may relate to a charged lysine in the middle of its TMD—unusual for transmembrane proteins—and the absence of a well-defined TMD-C anchor, typically found in high-confidence substrates. In summary, fuzzy labeling provides a refined explanation for uncertain predictions and enhances their interpretability.

To improve clarity in the main text, we revised the section on fuzzy labeling. We first introduced the concept, emphasizing how it addresses uncertainty in predictions, and then demonstrated its practical application using the example of TREM2. This structure reflects our discussions and is shown in the following excerpt.

Excerpt from the revised manuscript (main text, pages 10, 44–45):

(I) Explaining prediction results at the amino acid sequence level

... SHAP values depend on the machine learning model, the dataset, and how samples are labeled during training—*i.e.*, whether they are marked as positive or negative. When extending our SHAP analysis to proteins that are not included in the training dataset, the challenge arises of whether to label them as positive (*i.e.*, substrate) or negative (*i.e.*, non-substrate). To address this labeling ambiguity, we developed fuzzy labeling. In this novel procedure, machine learning models are trained over multiple

rounds, and the frequency of labeling proteins with unknown status (as positive or negative) is set corresponding to their prediction score. This probabilistic approach is designed for proteins that are absent from a training dataset and is particularly useful for explaining uncertain prediction scores reflected by high variance. We showcased fuzzy labeling for TREM2, which is a substrate from SUBLIT, but predicted as a low-confidence non-substrate ($36\pm 20\%$).

In dataset 1, where TREM2 is not included, its prediction score is 22%. However, labeling TREM2 as substrate yields a SHAP value sum of 0.66 (Extended Data Fig. 6d). Applying fuzzy labeling on TREM2 during 25 training rounds (*i.e.*, labeling it 5 rounds as substrate and 20 rounds as non-substrate) results in a SHAP value sum of 0.27, closely reflecting its 22% prediction score (Extended Data Fig. 6e). For dataset 1, the low prediction score of TREM2 is explained by the lack of disordered, large, and charged residues in the TMD-C anchor (Extended Data Fig. 6e-g). For dataset 2, which contains TREM2, it is always labeled as a substrate so that fuzzy labeling is not applied. This results in a higher prediction score of 62%, explained by positive-impact features overruling negative ones (Extended Data Fig. 6h). Basically, fuzzy labeling ensures that SHAP values reliably approximate prediction scores for any protein with unknown substrate status, which is especially useful for prediction scores with high variance. This approach allows for an in-depth comparison of physicochemical signatures, as illustrated for the validated HC substrate ERBB2 and LC non-substrate SLC27A1 (Extended Data Fig. 7a, b).

(II) Extended Data Fig. 6d-h

d–g, Fuzzy labeling demonstrated with TREM2 from SUBLIT (36±22% substrate prediction score) to explain its prediction score of 22% (blue) based on TMHMM annotation and dataset 1. Two labeling strategies for TREM2 over 25 rounds are **compared**: **(d)** consistently labeling it as a substrate; and **(e)** applying fuzzy labeling, where TREM2 is labeled 5 times as a substrate and 20 times as a non-substrate, corresponding to its prediction score. **Besides the SHAP force plot (e), the CPP-SHAP ranking plot (f) and CPP-SHAP profile (g)** illustrate the results of fuzzy labeling for TREM2. **h**, The CPP-SHAP profile **explaining** the TREM2 prediction score of 62% based on the TMHMM annotation and dataset 2, in which TREM2 is included (as member of SUBLIT) and therefore labeled as a substrate.

Comment 16: The work related to discovering substrate subgroups based on feature impact in the **"Explaining Prediction Results at the Amino Acid Sequence Level"** section seems **redundant**, as it is straightforward that similar feature embeddings lead to similar model outputs (i.e., substrate probability).

Response: We hope there was no confusion regarding the terminology used, as in the context of **Extended Data Fig. 8**, we refer to CPP features rather than embeddings. We appreciate the reviewer's observation regarding the connection between similar CPP features and similar model outputs. However, the aim of this analysis was not to confirm that similar CPP features result in similar substrate prediction scores. Instead, we sought to demonstrate how specific patterns of feature impact, as determined by the SHAP explainable AI technique, can be used to further identify groups of substrates characterized by a common physicochemical profile, which is reflected by correlating feature impacts between substrates.

This additional layer of interpretation provides valuable insights into the underlying biological meaning of the CPP features, going beyond what can be inferred from substrate prediction scores alone. Therefore, we believe that this analysis contributes to a deeper understanding of the substrate classifications and the distinct properties that define them, offering insights beyond the expected correlation between similar features and similar substrate prediction scores.

Reviewer #2:

This study employs AI to identify the substrate proteome of gamma-secretase. Although over 150 substrates have been identified to date, many additional substrates are believed to be at large due to the lack of sequence specificity: the enzyme appears to process the transmembrane domain of almost any N-out membrane protein with a short lumen/extracellular ectodomain (typically formed through a prior sheddase cleavage event). Only a handful of proteins have been identified as nonsubstrates, and the rules for what makes a substrate vs. a nonsubstrate (beyond a short N-terminal ectodomain) remain unknown. By providing a training set of known substrates, the authors were able to develop a highly predictive algorithm for substrates and nonsubstrates through machine learning, validating the approach through biological assays. Importantly, the researcher team evaluated RNA levels at tissue and single-cell levels to show that newly identified substrates are indeed co-expressed with gamma-secretase. **Although many new candidate substrates were identified, only a small fraction were tested biologically.** Nevertheless, this study provides rankings/probabilities for all substrates, allowing other investigators to determine if their membrane protein of interest might be worthwhile to test as an actual substrate. The authors also conducted bioinformatics analyses to connect the new substrates to potential biological and pathological roles.

The algorithm ranks physicochemical and conformational features that are commonly found in substrates. Most notable is the ability of the region around the initial cleavage site (in the C-terminal region of the transmembrane domain) to readily assume an alpha helix or a beta-strand. This is consistent

with the unbound transmembrane domain being helical (as seen by NMR for APP and Notch) but adjusting to a beta-strand in binding to the gamma-secretase active site (as seen in cryoEM structures of bound substrates).

Overall, this is an excellent study and represents a substantial advance in understanding the biology of gamma-secretase, an intramembrane protease complex central to metazoan biology and human disease.

We thank the reviewer for the appreciation of our work. Although not requested by this reviewer, we have included an analysis of additional data from a recently published proteomics screen of endogenously expressed γ -secretase substrates (Hou et al. 2023) to address comment 1 of Reviewer 1 (Fig. 4e). This extends the biological testing of predicted candidates, which was also mentioned above, in both quantity as well as quality, *i.e.*, by an analysis of endogenously expressed proteins in addition to overexpressed ones. This additional analysis further supports our study.

Appendix 1: Hardware specifications and computational time analysis

For our computational analysis, we utilized an Intel Core i7-10510U CPU (1.80 GHz, 4 cores, 8 threads) machine with 15 GB RAM, and Fedora 32 OS. Generating and filtering ~130,000 features (using default CPP feature settings with scale set 5) for comparing 63 substrates (SUBEXPERT) against 631 reference proteins (OTHERS) required ~7.5 minutes (see **Response Fig. 2**). A single training round (see **Extended Data Fig. 3a**)—including recursive feature selection, hyperparameter optimization using grid search (see **Supplementary Table 10** for parameters of all 10 models), and model evaluation—required ~5 minutes in total. Most of this time was spent on recursive feature selection (~1.5 minute) and training the tree-based models (30 second to 1 minute per model), while optimizing simpler models, such as support vector machine, took only ~1 second. Multi-processing allowed training the ensemble of 1500 models within 8 hours

Appendix 2: Performance evaluation on unseen data

To evaluate the performance of our substrate predictions on unseen data, we reanalyzed the results of the proteomics screen for endogenously expressed γ -secretase substrates in microglia-like cells (Hou et al., 2023). They used a \log_2 FC threshold of 0.3 as their identification criterion, identifying 59 proteins as new γ -secretase substrate candidates. However, we opted for a stricter \log_2 FC cut-off of 1, as a more permissive cut-off (e.g., 0.5) would incorrectly identify the known non-substrates TYROBP (\log_2 FC=0.61) and RELL1 (\log_2 FC=0.53) as substrates. Among the proteins with a \log_2 FC greater than or equal to 1 (48 in total), we selected only those not previously identified as γ -secretase substrates, resulting in a total of 28 proteins. Of these, 24 were predicted to be either low-confidence (LC) or high-confidence (HC) substrates, achieving a success rate of 86% on unseen data (see **Response Fig. 1, left**). The remaining 4 proteins, identified as γ -secretase substrates in the proteomics screen, were predicted as LC non-substrates.

For completeness, we also assessed the accuracy of prediction for known γ -secretase substrates. Among the 20 proteins with a \log_2 FC greater than or equal to 1, we correctly predicted 18 as LC or HC substrates (see **Response Fig. 1, right**).

Response Fig. 1 | Performance evaluation of substrate predictions against proteomics screen data for both new and known γ -secretase substrates. The new γ -secretase substrates (left panel) represent unseen data, while the known substrates (right panel) were included during training. For a detailed description of the scatterplots, see Comment 1 (excerpt from Manuscript III). The scatterplot in the manuscript (**Fig. 4e**) includes both known and new γ -secretase substrates.

Appendix 3: Optimization of CPP pre-filtering and filtering settings

Optimization of CPP pre-filtering settings

To assess the impact of different CPP pre-filtering settings, we tested combinations of the percentage of features retained after pre-filtered (*pct_pre_filter*: 5%, 10%, 20%) and the maximum allowed standard deviation in the test dataset (*max_std_test*: 0.05, 0.1, 0.2, 0.25, 0.3, 0.4). All other CPP parameters were kept at their default values, with SUBEXPERT as test set and OTHERS as reference set, annotated with THMMM. The prediction performance of random forest and support vector machine models (both with default settings) was evaluated using three approaches: (a) leave-one-out-cross-validation (LOOCV) with training on SUBEXPERT against NONSUB and NONSUBPRED, using only SUBEXPERT and NONSUB for validation (as in Fig. 2), with balanced accuracy as the performance measure (*loocv_BAC*); (b) 5-fold-cross-validation on a training set (80% of the total dataset containing SUBEXPERT, NONSUB, and NONSUBPRED), with the average accuracy over the 5 folds used as the performance measure (*cv_mean_ACC*); and (c) validation on a test set (20% of the total dataset) using accuracy for evaluation (*ACC*). In addition, we measured the time required for feature engineering (*i.e.*, creating and filtering of ~130.000 features) and recorded the number of final CPP features that were retained after the pre-filtering and filtering steps (Response Fig. 2).

The default setting (*pct_pre_filter*=5, *max_std_test*=0.2) demonstrated the most balanced and consistent performance across all metrics. For random forest models, these settings yielded the best LOOCV performance (85%). However, increasing *pct_pre_filter* led to an improvement from 83% to 87% for support vector machine models. In 5-fold-cross-validation, the default settings produced the highest accuracy for support vector machine models (89%) compared to 83–87% for higher *pct_pre_filter* values. Only minor differences were observed for random forest models, with performance ranging between 81% and 83%.

When evaluating accuracy on the test dataset, the default settings were very close to the best-performing configuration for both models, with most values near 100%. The only exception was the lowest *max_std_test* setting (0.05), which reduced random model performance down to 91%, likely due to strict filtering that retained only 53 features (compared to 100 for all other *max_std_test* settings).

Regarding CPP feature engineering time, higher *pct_pre_filter* values led to longer times, independent of *max_std_test* (~8.5 minutes for *pct_pre_filter*=10 and ~10.5 minutes for *pct_pre_filter*=20). Therefore, the default settings (including *pct_pre_filter*=5) offer an efficient pre-filtering process, while consistently optimizing prediction performance.

		CPP Pre-Filtering Settings																				
		5	10	20	5	10	20	5	10	20	5	10	20	5	10	20	5	10	20	5	10	20
		0.05			0.1			0.15			0.2			0.25			0.3			0.4		
		Evaluation Metrics for Random Forest (RF) and Support Vector Machine (SVM) models [%]																				
loocv_BAC (RF)		78	78	78	81	79	84	80	81	83	85	80	79	77	80	80	79	80	82	80	77	82
cv_mean_ACC (RF)		82	82	82	82	82	84	83	83	81	82	82	81	82	81	82	82	82	82	82	82	82
ACC (RF)		91	91	91	100	100	100	99	99	100	98	100	98	95	97	98	98	99	98	97	97	98
loocv_BAC (SVM)		78	78	78	81	85	81	82	81	83	83	87	87	83	83	87	79	83	87	83	83	87
cv_mean_ACC (SVM)		81	81	81	87	88	87	88	88	88	89	89	89	87	88	87	87	88	88	87	87	88
ACC (SVM)		100	100	100	100	100	96	96	96	96	100	100	96	100	100	96	100	100	100	100	100	100
		Time for CPP Feature Engineering [Seconds]																				
		447	439	436	454	514	625	449	509	634	446	513	630	444	505	629	446	518	628	450	506	624
		Number of CPP Features																				
		52	52	52	100	100	100	100	100	100	100	100	100	100	100	100	100	100	100	100	100	100

Response Fig. 2 | Optimization of CPP pre-filtering parameters *pct_pre_filter* and *max_std_test*. Heatmaps illustrating the impact of different CPP pre-filtering settings on the prediction performance, the feature generation time, and the number of features retained after the final filtering. The pre-filtering settings combine *pct_pre_filter* (5%, 10%, 20%) with varying *max_std_test* thresholds (0.05, 0.1, 0.15, 0.2, 0.25, 0.3, 0.4). The prediction performance of random forest (RF) and support vector machine (SVM) models was evaluated using leave-one-out-cross-validation with balanced accuracy (loocv_BAC), the average accuracy from 5-fold-cross-validation on a training dataset (cv_mean_ACC), and the accuracy on a test dataset (ACC). The time required for feature engineering (generating and filtering of features) and the number of features retained after filtering are shown. The default settings used in our analysis, along with their corresponding results, are highlighted by black squares.

Optimization of CPP filtering settings

To assess the impact of different CPP filtering settings, we tested combinations of the maximum allowed overlap between feature positions (*max_overlap*: 0, 0.25, 0.5, 0.75, 1) and the maximum allowed correlation between property scales used for features (*max_cor*: 0, 0.25, 0.5, 0.75, 1), while keeping all other CPP parameters at their default values. The prediction performance of random forest and support vector machine models was evaluated as before using leave-one-out cross-validation (loocv_BAC), 5-fold cross-validation on a training set (cv_mean_ACC), and accuracy on the test set (ACC). We also measured the feature engineering time and the number of final CPP features (**Response Fig. 3**).

The default setting (*max_overlap*=0.5, *max_cor*=0.5) provided the most balanced performance across all metrics. For random forest models, these settings achieved the highest LOOCV performance (85%), while support vector machine models performed best under stricter settings (*max_overlap*=0.25, *max_cor*=0.0), improving LOOCV balanced accuracy from 83% to 86%. Conversely, in 5-fold cross-validation, the default settings produced the highest average accuracy for support vector machine models (89%), whereas the stricter settings—that favoured support vector machine models in LOOCV—were best for random forest models (86% vs 81% for default settings). When evaluating accuracy on the test dataset, moderate to strict filtering settings for *max_overlap* (0.5 or 0.75) showed the best performance (96–100%) for both models, while very loose (*max_cor*=0) or strict (*max_cor*=1) correlation settings led to a performance drop to 90% or lower.

The time for CPP feature engineering remained stable across all configurations (~450 seconds). The number of retained CPP features varied with the filtering thresholds, where stricter settings (e.g., *max_overlap*=0, *max_cor*=0) significantly reduced features to as few as 17, likely contributing to the observed decrease in model performance. Overall, the default settings offer a robust trade-off between redundancy reduction and computational efficiency.

Response Fig. 3 | Optimization of CPP filtering parameters *max_overlap* and *max_cor*. Heatmaps illustrating the impact of different CPP filtering settings on the prediction performance, the feature engineering time, and the number of features retained after the final filtering. The filtering settings combine *max_overlap* and *max_cor*, both with the same threshold range (0, 0.25, 0.5, 0.75, 1). See **Response Fig. 2** for details on the prediction performance, the feature creation time, and the number of features. The default settings used in our analysis, along with their corresponding results, are highlighted by black squares.

Appendix 4: Robustness assessment of substrate and feature identification

To assess the robustness of our substrate prediction and feature identification approach, we assessed how varying the number of training rounds (**Extended Data Fig. 3a**) affects the reproducibility and consistency of prediction performance, feature importance scores, and prediction scores. We employed random forest models with default settings to ensure consistency with **Appendix 3**. As before, CPP features were generated with default values comparing SUBEXPERT (test set) against OTHERS (reference set) with THMMM TMD annotation. The detailed results are presented in **Response Fig. 4**.

We increased the number of training rounds from 1 to 25 to obtain Monte Carlo estimates of model predictions and feature importance scores by averaging over multiple rounds, aiming to enhance the robustness of our results as the number of training rounds increases. For each number of training rounds, we performed 10 iterations to assess variability. For example, with one training round, we trained and evaluated one model over 10 separate iterations (totaling 10 models) and recorded the average score and variance across all iterations. For two training rounds, we trained two models per iteration and computed their average evaluation metrics (accuracy, feature importance scores, and prediction scores), repeating this process over 10 iterations (totaling 20 models). This pattern continued up to 25 training rounds, where 25 models were trained per iteration, repeated over 10 iterations (250 models total). For each number of training rounds, we calculated the average accuracy, feature importance scores, and substrate prediction scores, along with their variances, across the 10 iterations.

To simplify visualization and interpretation, we chose the top 10 CPP features and further selected 5 of them to reduce overlapping variances. For prediction scores, we randomly sampled 10 proteins from OTHERS that were predicted as low-confidence substrates (*i.e.*, final substrate prediction score between 50% and 80%) to ensure comparability. From these, 5 proteins were selected again to minimize overlapping variance, providing a clearer view of trends.

The **top panel of Response Fig. 4** illustrates the average cross-validation accuracy (*cv_mean_ACC*) of the random forest model over 10 iterations, plotted against the number of training rounds. The shaded area represents the variance observed across iterations. The results show that the average accuracy remains stable, fluctuating between 82% and 83%. Notably, as the number of rounds increased, the variance decreased, indicating an improvement in the stability and reliability of the model's performance.

The **middle and bottom panels of Response Fig. 4** present the feature importance scores and the prediction scores for the five selected features and proteins, respectively, across the 25 training rounds. The middle panel shows that while feature importance values fluctuate, they remained generally stable and show lower variance with increasing training rounds, suggesting enhanced reproducibility. Similarly, the bottom panel demonstrates that prediction scores for the selected proteins are consistent over the rounds, with only slight variations. This stability underscores that increasing the number of training rounds strengthens the consistency of the substrate identification and feature ranking.

Response Fig. 4 | Impact of number of training rounds on model performance, feature importance, and prediction scores. Line plots showing the average cross-validation accuracy (cv_mean_ACC), feature importance scores, and prediction scores over 10 iterations for a random forest model, plotted across 1 to 25 training rounds. The shaded area indicates the variance (standard deviation).

Point to point response to new comments

Reviewer #3:

New comment 1: Benchmarking of dPULearn Against Classical PU Methods (Elkanoto)

It is unclear how dPULearn is benchmarked against Elkanoto methods. The authors do not mention any external test set on which the classification task was performed. If we understand correctly, stating that the classifier performs better on a dataset with true positives and predicted negatives does not necessarily confirm that the original labeling method was superior. For instance, if distant proteins were selected as negatives via PCA, this could result in two distinct clusters that are easy to classify, without necessarily confirming that the selected proteins were truly negative (or were useful negatives as training data for the classification task). While this does not entirely call into question the validity of the results, further clarification on this aspect would be helpful.

Response (new): We thank the reviewer for his thorough evaluation and have added a reference in the main manuscript to the *Supplementary Information* (Benchmarking dPULearn), where the full benchmarking details are provided. The corresponding revision is highlighted in red in the manuscript, as shown in the excerpt below.

We fully agree that evaluating a classifier solely based on predicted negatives rather than known true negatives is insufficient. This concern was already addressed in our original manuscript through two complementary evaluation approaches: (1) assessing performance using known non-substrates together with newly identified non-substrates, and (2) testing generalization by evaluating model predictions only on known non-substrates. The second strategy directly aligns with the reviewer's concern and ensure a fair assessment of dPULearn's performance. Both strategies are summarized in the following.

To systematically benchmark dPULearn against Elkanoto PU methods, we employed random forest and support vector machine classifiers with leave-one-out cross-validation. As first approach, we evaluated whether the models could effectively learn from the newly identified non-substrates, using datasets consisting of all known substrates (SUBEXPERT) and known non-substrates (NONSUB) alongside the newly identified non-substrates for each method (**Extended Data Fig. 2f, left**). In the second approach, we assessed generalization performance by computing the true negative rate (TNR), evaluating model predictions only on known non-substrates after training on SUBEXPERT and the newly identified non-substrates (**Extended Data Fig. 2f, right**). This approach ensured that model performance was not biased by dataset artifacts or trivial separability but reflected a meaningful ability to distinguish non-substrates. dPULearn performed close to the best models in both evaluation approaches, demonstrating a strong balance between predictive accuracy and generalization.

To further confirm robustness, we tested the effect of parameter variations on identified non-substrates (**Extended Data Fig. 2h–j**) and assessed reproducibility across multiple runs (**Extended Data Fig. 2g**), showing that dPULearn consistently produced stable and repeatable results. Additionally, we compared its performance to models trained on randomly sampled non-substrates, confirming a statistically significant improvement over random selection ($P < 0.001$, **Extended Data Fig. 2k**). These findings demonstrate that dPULearn is both reliable and generalizable, making it a robust alternative to classical Elkanoto PU learning methods.

Excerpt from the revised manuscript (main text, page 4):

We benchmarked dPULearn (*Supplementary Information*) against the popular PU learning framework developed by Elkan and Noto⁴⁰ (referred to as Elkanoto), which uses machine learning classification models and is, therefore, a stochastic approach....

New comment 2: Concerns Regarding Overfitting and Model Evaluation Metrics

The authors tested the model on a new proteomic dataset to address concerns about potential overfitting and reported a “success rate” of 88%. However, if I understand correctly, this metric appears to be based on recall. A naive model predicting only positives would achieve 100% recall, which does not necessarily indicate good predictive performance. Additionally, to properly assess overfitting, the same evaluation metric should be used for both training and testing. The authors mention that the training and validation metric was balanced accuracy, yet they compare it to recall when evaluating the new dataset. **Could the authors clarify why different metrics were used?** Based on accuracy, calculated as (true positives + true negatives) / total, the general performance appears to be closer to 73%, indicating potential overfitting. It also should be put into context by comparing to a random classifier, which would achieve around 50.

Response (new): We thank the reviewer for his comment and appreciate the opportunity to clarify our evaluation approach.

For the proteomic screen by Hou et al. (2023), which we used for further validation on unseen data, only identified substrates are provided—there is no validated set of non-substrates. Whether a protein is experimentally validated as a substrate on such a screen depends on the chosen detection threshold. We deliberately applied a stricter threshold, which aligns with standard practice in the field, to ensure that only high-confidence substrates were considered.

The reviewer’s calculation assumes that all proteins below this threshold should be classified as true non-substrates, which is incorrect. The absence of detection on the screen does not confirm that a protein is a true non-substrate, but rather that it remains inconclusive. In fact, in the original publication of the proteomic dataset, these proteins were classified as substrate candidates, further demonstrating that their status cannot be assumed as negative. Consequently, computing true negatives based on these assumptions would be misleading and methodologically inappropriate.

Regarding the concern about the naive model assumption, our predictor has been rigorously evaluated both experimentally and computationally, consistently demonstrating a strong predictive performance of ~90%. This is far from a naive model that simply predicts all positives. Our approach has been validated through cross-validation strategies, confirming its robustness and reliability.

We intentionally used the term “success rate” rather than “recall” to clearly distinguish this experimental validation, which is based on substrates identified in a literature-derived proteomic screen, from our curated machine learning datasets, which include both validated substrates and non-substrates confirmed through biochemical assays.

We also note that while the assumption that a random classifier would achieve 50% accuracy is computationally correct, it is biologically questionable, as this would only be valid if substrates constituted 50% of the human N-out membrane proteome. Our study shows that this proportion is likely between 20% and 50%, depending on the substrate prediction threshold. However, this does not affect our core argument, as our model’s performance remains well above any realistic random baseline approach.

Please note, that we had already described in the revised manuscript that the Hou et al. (2023) dataset only contains substrates (“... we used a recent proteomics screen that identified 85 substrate candidates in human microglia-like cells by pharmacological γ -secretase inhibition⁴⁹”).

Reviewer #1 (with Re-Review of Reviewer #3):

Comment 1: As for the machine learning workflow for γ -secretase substrate prediction, the predictions in this study are based on aggregated scores from 1,500 trained models. However, these models are

trained on a small dataset consisting of only a few hundred substrate samples. Given the limited data, **the computational cost of building such an ensemble might outweigh its benefits**. There is also a **significant risk that the ensemble could collectively overfit to the small dataset, resulting in good performance on the training data but poor generalization to unseen data**.

Response: Given the relatively small size of the training dataset, the computational cost of training the ensemble of models was minimal (see **Appendix 1** for detailed hardware specifications and computation time). This ensemble approach was chosen to ensure reproducibility and robustness in the results (see **answer to comment 7** including **Appendix 4** for further results and discussion).

While we acknowledge the limited size of our dataset, which increases the risk of overfitting, we have already implemented measures to mitigate this risk. Despite the small number of predicted substrates (n=12) and non-substrates (n=5) that were experimentally tested and validated with a success rate of 88%, our model showed a strong generalization on these unseen data, aligning well with computational accuracy of 90% (**Fig. 2**). To further assess generalization on unseen data, we now additionally validated our approach using an independent proteomic screening of endogenously expressed γ -secretase substrates from the literature (Hou et al., 2023). Using a stricter identification criterion (see **Appendix 2**), we correctly predicted 42 out of 48 substrates, resulting also in an 88% success rate. These 48 proteins included 28 proteins for which it was not known whether they are substrates or non-substrates, *i.e.*, unseen data points. 24 of these 28 proteins were predicted to be substrates, yielding a similar success rate of 86%. This independent validation underscores the model's ability to generalize beyond the initial training data. To illustrate the correlation between our predictions and the proteomics results, we revised two sections of the main text and added a new panel figure (**Fig. 4e**), as shown in the following excerpts: [...]

Re-Review 1: By providing both empirical validation and manuscript updates, the response effectively mitigates concerns about overfitting. However, the evaluation approach raises potential bias concerns. Since the model's performance is assessed primarily on experimentally identified substrates (Fig. 4e), there is a risk of overly optimistic prediction. A naive model that always predicts "substrate" could appear highly accurate under such conditions (using recall). A more rigorous evaluation should compare predictions against a balanced dataset that includes both true substrates and true non-substrates, ensuring that performance metrics are properly assessed and that the model generalizes well.

The overall accuracy based on figure 4.e should be: $(\text{True Positives} + \text{True Negatives}) / \text{Total}$, which leads to around 73%. The positive accuracy should be: $\text{True Positives} / (\text{True Positives} + \text{False Negatives})$, which leads to around 72%. And the negative accuracy: $\text{True Negatives} / (\text{True Negatives} + \text{False Negatives})$, which leads to around 75%.

This should be compared with a random predictor, which would achieve around 50% accuracy on a balanced dataset. While this performance is better than random, the authors should moderate their claims regarding the model's performance on this proteomic test set. (Additionally, the trade-off between ensemble complexity and dataset size could have been better addressed by comparing the ensemble approach with simpler models. These additional performance metrics would provide a more transparent assessment of the model's real-world predictive power.)

Response (new): Please refer to our response to **New Comment 2** for clarification on this point. Regarding the additional request to compare the ensemble approach with simpler models, we address this in **Comment 7** and provide further details in **Appendix 4**.

In the interest of readability and clarity of our manuscript, we prefer not to include the additional benchmarking of **Appendix 4** into our manuscript. Additionally, our model has undergone rigorous experimental validation across multiple independent in vitro systems, including screening data from the literature, providing a strong indication of its predictive reliability in real-world applications.

Comment 4: In the section on Identification of additional non-substrates by dPULearn, the authors mention, “For each PC, proteins from OTHERS that are most distant to SUBEXPERT proteins are identified as additional non-substrates.” **How is the distance quantified, and how was the number of predicted non-substrates (NONSUBPRED) set to 49?**

Response: The distance between proteins from OTHERS and SUBEXPERT proteins is quantified using the absolute differences in their principal component (PC) values. dPULearn applies principal component analysis (PCA) to reduce the high-dimensional feature space onto principal components, each representing a dimension of variance. For each PC, the distance between a protein from OTHERS and the mean PC value of all SUBEXPERT proteins is calculated. Proteins from OTHERS with the greatest absolute distances from the SUBEXPERT mean for the respective PC (e.g., the mean for PC1) are selected as additional non-substrates.

The number of predicted non-substrates (NONSUBPRED) was set to 49 to balance the dataset. Given that there are 63 expert-curated substrates (SUBEXPERT) and 14 known non-substrates (NONSUB), we identified 49 additional non-substrates (NONSUBPRED) to equalize the total number of non-substrates and substrates at 63 each. This ensures a balanced dataset for training the machine learning model.

For further details on the algorithm and selection process, please refer to the "Computational non-substrate identification by dPULearn" section of the *Supplementary Information*. To enhance the clarity of our manuscript, we have incorporated the answers to this comment into the main text, as shown in the following excerpt: [...]

Re-Review 4: The response clearly explains how distance is measured using absolute PC differences and justifies selecting 49 non-substrates to balance the dataset. **However, it does not address potential bias in dataset balancing, selecting non-substrates solely based on PCA distance may over-represent outliers rather than true non-substrates.**

Response (new): Selecting non-substrates based on lower principal components (PCs), i.e., those with lower explained variance, does carry the risk of including outliers—non-substrates that might actually be true substrates. However, selecting non-substrates only from the first PC introduces a different bias, as it favours non-substrates that are far from the decision boundary and focuses on a limited feature subspace. This could lead to a training set that does not adequately represent the full variability of the dataset (see **Extended Data Fig. 2**).

To balance these risks, dPULearn selects non-substrates based on a weighted approach, considering the relative amount of explained variance and limiting the number of top PCs used in the selection process. This ensures a representative set of potential non-substrates while minimizing the risk of outliers.

Our results confirm that the selected non-substrates provide a reliable dataset for training. As shown in Fig. 3f, only 2 out of the 49 selected non-substrates (CLMP, ATRN) have a substrate prediction score above 50%, classifying them as low-confidence substrates. In contrast, 20 have prediction scores below 20%, indicating high-confidence non-substrates. This suggests that the risk of misclassification, on our dataset, is low (~4%) using dPULearn, supporting its robustness in non-substrate selection.

That said, the reviewer raises a valid concern: if too many negatives are identified or if a large number of PCs with low explained variance are used (indicating a highly diverse feature space with few correlating features), dPULearn could potentially select more outliers. To address this, we recommend that users set the number of PCs based on a fixed count rather than a percentage of explained variance (by providing an integer value ≥ 1 as input for ‘n_components’). We have added a corresponding warning

in the dPULearn documentation within our AAanalysis Python package to ensure users are aware of this limitation. Investigating this issue across diverse datasets is an important direction for future studies, but this is beyond the scope of this manuscript.

Comment 7: In the Section of Feature ranking using machine learning, the authors selected the 6 best approaches for different dataset-annotation combinations, resulting in 1,500=6×250 trained models. **Have the authors investigated how the number of models in the ensemble affects performance? Is such a large ensemble essential for substrate identification?**

Response: We continuously assessed how the number of models in the ensemble impacted prediction performance, substrate prediction score robustness, and feature importance throughout the project. Our primary goal was to ensure reproducibility, which we achieved through iterative improvements. As described in the “Learning strategy” section of the *Methods* part, we performed 25 independent training rounds, each involving dataset splits, feature selection, hyperparameter optimization, and model evaluation (**Extended Data Fig. 3a**). This provided a Monte Carlo estimate of prediction scores and the feature importance, ensuring robustness in our results.

Additionally, we incorporated different datasets and annotations to estimate uncertainty arising from biological variability. By training models on different TMD annotations (representing different biological assumptions), we gained a more comprehensive understanding of how these variations might affect substrate predictions, resulting in more reliable and generalizable outcomes.

While these optimizations were crucial to improving robustness, we chose not to include detailed analyses of how robustness scales with the number of training rounds and models in the main text to avoid excessive technical details. However, we now conducted three in-silico experiments to examine the effects of the number of models on the (a) prediction performance, (b) reproducibility of feature importance, and (c) consistency of the substrate prediction score. These findings are detailed in **Appendix 4**.

Re-Review: The response effectively explains that the number of models was assessed for its impact on prediction performance, robustness, and feature importance. However, it does not clearly state whether a smaller ensemble could achieve similar performance, which was the core of the reviewer’s question. I also was unable to find Appendix 4.

Response: (new) We apologize for the confusion regarding Appendix 4—it was included at the end of our previous response letter rather than in the manuscript or *Supplementary Information*. To address this, we have now appended Appendix 4 again at the end of this response letter.

Regarding the core question of whether a smaller ensemble could achieve similar performance: Yes, our analyses confirm that beyond a certain number of models, additional training rounds do not alter overall accuracy, which remains within the standard deviation range (82%–83%). However, increasing the ensemble size enhances robustness by reducing variance in feature importance and prediction scores, leading to more reproducible results. As detailed in **Appendix 4**, while a smaller ensemble achieves comparable accuracy, a larger ensemble improves stability and reliability.

Appendix 4 provides a full breakdown of these results, demonstrating how prediction performance, feature reproducibility, and substrate prediction score consistency scale with ensemble size.

Comment 13: In the section CPP and dPULearn outperform state-of-the-art methods, the authors use support vector machine models with default settings to conduct a model performance comparison. **A**

fairer comparison would involve using the same model architecture as in this study instead of the default support vector machine models.

Response: We appreciate the reviewer’s suggestion and agree that using the same model architecture could offer an interesting alternative perspective. However, our primary goal in this comparison was to assess the general utility of different feature engineering approaches (scale-based, embeddings, and CPP) in a standardized machine learning setting. By using support vector machine (SVM) models with default settings, we ensured the same basic conditions where the performance differences could be attributed primarily to the feature engineering techniques rather than differences in model optimization or architecture. This allowed us to focus on how effectively each method translates raw data into meaningful features for prediction.

Additionally, SVM is a well-established baseline for binary classification, particularly suited to small datasets like ours, where overfitting is a concern. The default settings ensured that results weren’t overly influenced by hyperparameter tuning, which would vary across model architectures.

While we plan to explore the use of different models in future work, introducing them here would add unnecessary complexity and be out of the scope of this work. We believe that focusing on feature engineering with a standard SVM model provides a clear and interpretable comparison of the methods.

To reflect this discussion, we modified the current version of the manuscript, as shown in the following excerpt: [...]

Re-Review: The response provides a justification for using SVM with default settings to isolate the impact of feature engineering techniques while avoiding confounding factors like model optimization. However, I have concerns regarding the benchmarking of dPULearn against Elkanoto methods, particularly the absence of an external test set (see summary statement).

Response (new): See answer to **New comment 1**.

Appendix 4: Robustness assessment of substrate and feature identification

To assess the robustness of our substrate prediction and feature identification approach, we assessed how varying the number of training rounds (**Extended Data Fig. 3a**) affects the reproducibility and consistency of prediction performance, feature importance scores, and prediction scores. We employed random forest models with default settings to ensure consistency with **Appendix 3**. As before, CPP features were generated with default values comparing SUBEXPERT (test set) against OTHERS (reference set) with THMM TMD annotation. The detailed results are presented in **Response Fig. 4**.

We increased the number of training rounds from 1 to 25 to obtain Monte Carlo estimates of model predictions and feature importance scores by averaging over multiple rounds, aiming to enhance the robustness of our results as the number of training rounds increases. For each number of training rounds, we performed 10 iterations to assess variability. For example, with one training round, we trained and evaluated one model over 10 separate iterations (totaling 10 models) and recorded the average score and variance across all iterations. For two training rounds, we trained two models per iteration and computed their average evaluation metrics (accuracy, feature importance scores, and prediction scores), repeating this process over 10 iterations (totaling 20 models). This pattern continued up to 25 training rounds, where 25 models were trained per iteration, repeated over 10 iterations (250 models total). For each number of training rounds, we calculated the average accuracy, feature importance scores, and substrate prediction scores, along with their variances, across the 10 iterations.

To simplify visualization and interpretation, we chose the top 10 CPP features and further selected 5 of them to reduce overlapping variances. For prediction scores, we randomly sampled 10 proteins from OTHERS that were predicted as low-confidence substrates (*i.e.*, final substrate prediction score between 50% and 80%) to ensure comparability. From these, 5 proteins were selected again to minimize overlapping variance, providing a clearer view of trends.

The **top panel of Response Fig. 4** illustrates the average cross-validation accuracy (cv_mean_ACC) of the random forest model over 10 iterations, plotted against the number of training rounds. The shaded area represents the variance observed across iterations. The results show that the average accuracy remains stable, fluctuating between 82% and 83%. Notably, as the number of rounds increased, the variance decreased, indicating an improvement in the stability and reliability of the model's performance.

The **middle and bottom panels of Response Fig. 4** present the feature importance scores and the prediction scores for the five selected features and proteins, respectively, across the 25 training rounds. The middle panel shows that while feature importance values fluctuate, they remained generally stable and show lower variance with increasing training rounds, suggesting enhanced reproducibility. Similarly, the bottom panel demonstrates that prediction scores for the selected proteins are consistent over the rounds, with only slight variations. This stability underscores that increasing the number of training rounds strengthens the consistency of the substrate identification and feature ranking.

Response Fig. 4 | Impact of number of training rounds on model performance, feature importance, and prediction scores. Line plots showing the average cross-validation accuracy (cv_mean_ACC), feature importance scores, and prediction scores over 10 iterations for a random forest model, plotted across 1 to 25 training rounds. The shaded area indicates the variance (standard deviation).

Summary of the Re-Review

Overall, the authors have addressed most of the points raised by Reviewer 1. However, I still have a few concerns on which clarification would be helpful:

New comment: Benchmarking of dPULearn Against Classical PU Methods (Elkanoto)

It is unclear how dPULearn is benchmarked against Elkanoto methods. The authors do not mention any external test set on which the classification task was performed.

If we understand correctly, stating that the classifier performs better on a dataset with true positives and predicted negatives does not necessarily confirm that the original labeling method was superior. For instance, if distant proteins were selected as negatives via PCA, this could result in two distinct clusters that are easy to classify, without necessarily confirming that the selected proteins were truly negative (or were useful negatives as training data for the classification task). While this does not entirely call into question the validity of the results, further clarification on this aspect would be helpful.

Concerns Regarding Overfitting and Model Evaluation Metrics

The authors tested the model on a new proteomic dataset to address concerns about potential overfitting and reported a “success rate” of 88%. However, if I understand correctly, this metric appears to be based on recall. A naive model predicting only positives would achieve 100% recall, which does not necessarily indicate good predictive performance. Additionally, to properly assess overfitting, the same evaluation metric should be used for both training and testing. The authors mention that the training and validation metric was balanced accuracy, yet they compare it to recall when evaluating the new dataset. Could the authors clarify why different metrics were used?

Based on accuracy, calculated as $(\text{true positives} + \text{true negatives}) / \text{total}$, the general performance appears to be closer to 73%, indicating potential overfitting. It also should be put into context by comparing to a random classifier, which would achieve around 50%.

The author's previous point-by-point response to Reviewer #1's comments, together with our evaluation of those responses

Comment 1: As for the machine learning workflow for γ -secretase substrate prediction, the predictions in this study are based on aggregated scores from 1,500 trained models. However, these models are trained on a small dataset consisting of only a few hundred substrate samples. Given the limited data, **the computational cost of building such an ensemble might outweigh its benefits**. There is also a **significant risk that the ensemble could collectively overfit to the small dataset, resulting in good performance on the training data but poor generalization to unseen data**.

Response: Given the relatively small size of the training dataset, the computational cost of training the ensemble of models was minimal (see **Appendix 1** for detailed hardware specifications and computation time). This ensemble approach was chosen to ensure reproducibility and robustness in the results (see **answer to comment 7** including **Appendix 4** for further results and discussion).

While we acknowledge the limited size of our dataset, which increases the risk of overfitting, we have already implemented measures to mitigate this risk. Despite the small number of predicted substrates ($n=12$) and non-substrates ($n=5$) that were experimentally tested and validated with a success rate of 88%, our model showed a strong generalization on these unseen data, aligning well with computational accuracy of 90% (**Fig. 2**). To further assess generalization on unseen data, we now additionally validated our approach using an independent proteomic screening of endogenously expressed γ -secretase substrates from the literature (Hou et al., 2023). Using a stricter identification criterion (see **Appendix 2**), we correctly predicted 42 out of 48 substrates, resulting also in an 88% success rate. These 48 proteins included 28 proteins for which it was not known whether they are substrates or non-substrates, *i.e.*, unseen data points. 24 of these 28 proteins were predicted to be substrates, yielding a similar success rate of 86%. This independent validation underscores the model's ability to generalize beyond the initial training data. To illustrate the correlation between our predictions and the proteomics results, we revised two sections of the main text and added a new panel figure (**Fig. 4e**), as shown in the following excerpts: [...]

Re-Review:

By providing both empirical validation and manuscript updates, the response effectively mitigates concerns about overfitting. However, the evaluation approach raises potential bias concerns. Since the model's performance is assessed primarily on experimentally identified substrates (Fig. 4e), there is a risk of overly optimistic prediction. A naive model that always predicts "substrate" could appear highly accurate under such conditions (using recall). A more rigorous evaluation should compare predictions against a balanced dataset that includes both true substrates and true non-substrates, ensuring that performance metrics are properly assessed and that the model generalizes well.

The overall accuracy based on figure 4.e should be: $(\text{True Positives} + \text{True Negatives}) / \text{Total}$, which leads to around 73%.

The positive accuracy should be: $\text{True Positives}/(\text{True Positives}+\text{False Negatives})$, which leads to around 72%.

And the negative accuracy: $\text{True Negatives}/(\text{True Negatives}+\text{False Negatives})$, which leads to around 75%.

This should be compared with a random predictor, which would achieve around 50% accuracy on a balanced dataset. While this performance is better than random, the authors should moderate their claims regarding the model's performance on this proteomic test set.

(Additionally, the trade-off between ensemble complexity and dataset size could have been better addressed by comparing the ensemble approach with simpler models. These additional performance metrics would provide a more transparent assessment of the model's real-world predictive power.)

Comment 2: In the section on feature engineering using CPP, the authors mention that "sequence parts—in this case, the TMD and its adjacent N- and C-terminal juxtamembrane domains (JMD-N and JMD-C) of single-span membrane proteins—can be split into either continuous segments or discontinuous patterns, reflecting helical periodicity (Extended Data Fig. 1b–d)." **The rationale behind this splitting of sequence parts to reflect helical periodicity is unclear.** Please clarify the reasoning for this operation and **provide further details on the "Comparative Physicochemical Profiling (CPP)" feature.**

Response: The rationale for splitting sequence parts into discontinuous patterns is to capture general helical periodicity, where residues spaced 3/4 amino acids apart typically align on the same side of the helix. These residues form key interaction interfaces and are potentially crucial for substrate recognition since all γ -secretase substrates are single-span membrane proteins with a helical transmembrane domain. Our CPP approach uses this biologically inspired heuristic to balance capturing meaningful structural patterns while avoiding a combinatorial explosion. Limiting distances to 3 and 4 residues yields 182 patterns (see **Extended Data Fig. 1d** and **Supplementary Table 5**), whereas including distances of 1–4 residues would create over 5,000 patterns. Distances of 3 to 4 residues can represent helical face interactions, short-range interactions in unstructured regions, or short-range interactions that allow transitions between extended and helical conformations.

CPP features are defined by part-split-scale combination, as explained in detail in the "Splitting of sequence parts" section of the *Supplementary Information*. Examples of the positions of the residues resulting from part-split combinations are generally highlighted in our CPP feature ranking plots (in grey), such as in **Figure 5 a–c** and in **Extended Data Figure 6c**.

Since CPP features are introduced in detail in the *Supplementary Information* and **Extended Data Fig. 1**, we added a cross-reference to this section in the main text to enhance clarity. Additionally, we included the following revisions to the main text and Methods section to clarify the concept of periodic patterns reflecting helical periodicity.

Re-Review:

The response explains the reasoning behind the 3/4 residue splitting well and acknowledges the balance between capturing structure and avoiding complexity.

Comment 3: In the Methods section, the authors state, “To efficiently pre-filter these features, $max_std_test=0.2$ and $pct_pre_filter=0.05$ were used, yielding 6,583 features. In the filtering step, a value of 0.5 was empirically chosen for $max_overlap$ and max_cor .” Please **provide an explanation and justification for these threshold parameters**.

Response: We appreciate the reviewer’s request for clarification regarding the thresholds of the parameters used in the pre-filtering (max_std_test , pct_pre_filter) and filtering ($max_overlap$, max_cor) steps of the CPP algorithm. Below, we provide detailed explanations and justifications for these parameters:

- **max_std_test:** This parameter defines the maximum allowed standard deviation within the test dataset for feature pre-filtering. By setting $max_std_test=0.2$, we ensure that only features with low variability (maximum of 20%) within the test group are retained, reducing the risk of overfitting to noise by eliminating highly variable features that may lead to poor generalization.
- **pct_pre_filter:** This parameter specifies the percentage of features to retain after the pre-filtering step. We set $pct_pre_filter=0.05$ to retain the top 5% of features that exhibit the greatest discrimination between the test and reference groups (measured by absolute AUC), helping to focus the model on the most relevant features and improve the efficiency of the CPP algorithm. If the difference between the train and test datasets are weaker, this value may need to be increased.
- **max_overlap:** This parameter sets the maximum allowed positional overlap between features. With $max_overlap=0.5$, we limit redundancy by preventing the retention of features that overlap by more than 50% in their residue positions. A higher value would result in less strict filtering, increasing the number of retained features but also potentially introducing more redundant information.
- **max_cor:** This parameter defines the maximum allowed Pearson correlation between the property scales used for the features. Setting $max_cor=0.5$ ensures that features with high Pearson correlation (over 50%) regarding their used property scales are filtered out, maintaining only complementary and non-redundant features. Higher values would result in less strict filtering, leading to increased redundancy in the property scales used by CPP features.

The effect of different thresholds is demonstrated the CPP documentation and tutorial within the AAanalysis framework. These parameters, along with the effects of varying them, are detailed in the "CPP Algorithm" section of the *Supplementary Information*. We have also

added a remark in the “CPP algorithm” *Method* section of the main text explaining the rationale for choosing *max_overlap* and *max_cor* at 0.5 (50%), as a balanced trade-off between avoiding high redundancy and preventing the removal of too many complementary features (see following excerpt). We additionally performed an analysis on the impact of varying CPP pre-filtering (*pct_pre_filter* and *max_std_test*) and filtering (*max_overlap* and *max_cor*) setting thresholds on the prediction performance, the processing time, and the number of obtained CPP features (see **Appendix 3**).

Re-review:

The response provides a clear explanation of the parameters and their roles in feature selection, effectively addressing the reviewer’s concern. While a brief justification for the specific threshold choices could strengthen the response, the added remarks in the Methods section and Appendix 3 sufficiently clarify the rationale. This should be adequate.

Comment 4: In the section on Identification of additional non-substrates by dPULearn, the authors mention, “For each PC, proteins from OTHERS that are most distant to SUBEXPERT proteins are identified as additional non-substrates.” **How is the distance quantified, and how was the number of predicted non-substrates (NONSUBPRED) set to 49?**

Response: The distance between proteins from OTHERS and SUBEXPERT proteins is quantified using the absolute differences in their principal component (PC) values. dPULearn applies principal component analysis (PCA) to reduce the high-dimensional feature space onto principal components, each representing a dimension of variance. For each PC, the distance between a protein from OTHERS and the mean PC value of all SUBEXPERT proteins is calculated. Proteins from OTHERS with the greatest absolute distances from the SUBEXPERT mean for the respective PC (e.g., the mean for PC1) are selected as additional non-substrates.

The number of predicted non-substrates (NONSUBPRED) was set to 49 to balance the dataset. Given that there are 63 expert-curated substrates (SUBEXPERT) and 14 known non-substrates (NONSUB), we identified 49 additional non-substrates (NONSUBPRED) to equalize the total number of non-substrates and substrates at 63 each. This ensures a balanced dataset for training the machine learning model.

For further details on the algorithm and selection process, please refer to the “Computational non-substrate identification by dPULearn” section of the *Supplementary Information*. To enhance the clarity of our manuscript, we have incorporated the answers to this comment into the main text, as shown in the following excerpt.

Re-review:

The response clearly explains how distance is measured using absolute PC differences and justifies selecting 49 non-substrates to balance the dataset. However, it does not address potential bias in dataset balancing, selecting non-substrates solely based on PCA distance may over-represent outliers rather than true non-substrates.

Comment 5: In the same section, the authors state, “CPP achieved 84% balanced accuracy with an optimized part set and 92% accuracy when NONSUBPRED was included to balance the datasets.” **Is the reported model accuracy calculated with or without the inclusion of NONSUBPRED in the dataset?**

Response: We appreciate this important question and would like to clarify any confusion. The reported model performance of 84% balanced accuracy (**Extended Data Fig. 2l**) refers to the model trained and validated without the inclusion of NONSUBPRED, using only NONSUB and SUBEXPERT data. The 92% accuracy (**Extended Data Fig. 2m**) was achieved after NONSUBPRED was included to balance the dataset. NONSUBPRED was used for both training and evaluation, following the leave-one-out cross-validation approach, as detailed in the “Model optimization and evaluation” section of the *Supplementary Information*.

One of our key results, demonstrated in **Fig. 2a**, shows that including NONSUBPRED during training also improves performance when validated solely on NONSUB and SUBEXPERT. This result is described in the “CPP and dPULearn outperform state-of-the-art methods” section of the main text. Since this outcome appears later in the manuscript (page 7), we chose not to cross-reference it to **Extended Data Fig. 2m** (cross-referenced in page 5) to maintain clarity and flow in the earlier sections.

Re-review:

The response clarifies that the 84% balanced accuracy was achieved without NONSUBPRED, while the 92% accuracy was obtained after including NONSUBPRED to balance the dataset.

Comment 6: In the section on Identification of additional non-substrates by dPULearn, the manuscript states, “10 different machine learning classification model types—4 tree-based, 2 linear, 1 kernel-based, 1 neural network, and 2 ensemble model classes—were used in 25 training rounds, yielding 250=25×10 trained models.” The authors should **clarify whether each model contributed equally to the overall performance in substrate identification**. Validation is necessary to illustrate the effectiveness of each model.

Response: We appreciate this important question. Each of the 10 machine learning model types contributed equally to the overall performance in substrate identification. The prediction scores from all models were directly averaged without weighting across the model types or the 25 training rounds, as indicated in **Extended Data Fig. 3a**. While there were slight variations in performance depending on the dataset and annotation, all models showed robust performance (**Extended Data Fig. 3f**). Among the models, the multi-layer perceptron (MLP) showed slightly weaker performance compared to the others. However, overall, the inclusion of diverse model types contributed to balanced and robust predictions. For identifying the feature importance, we utilized the 4 tree-based models and their permutation-based feature importance.

We hope this clarifies the reviewer's question and believe that no further clarification in the main text is necessary. The fact that we computed an unweighted average across all models is already described in the manuscript, as shown in the following excerpt.

Re-review:

The response clarifies that all models contributed equally through unweighted averaging and acknowledges slight performance differences.

Comment 7: In the Section of Feature ranking using machine learning, the authors selected the 6 best approaches for different dataset-annotation combinations, resulting in 1,500=6×250 trained models. **Have the authors investigated how the number of models in the ensemble affects performance? Is such a large ensemble essential for substrate identification?**

Response: We continuously assessed how the number of models in the ensemble impacted prediction performance, substrate prediction score robustness, and feature importance throughout the project. Our primary goal was to ensure reproducibility, which we achieved through iterative improvements. As described in the “Learning strategy” section of the *Methods* part, we performed 25 independent training rounds, each involving dataset splits, feature selection, hyperparameter optimization, and model evaluation (**Extended Data Fig. 3a**). This provided a Monte Carlo estimate of prediction scores and the feature importance, ensuring robustness in our results.

Additionally, we incorporated different datasets and annotations to estimate uncertainty arising from biological variability. By training models on different TMD annotations (representing different biological assumptions), we gained a more comprehensive understanding of how these variations might affect substrate predictions, resulting in more reliable and generalizable outcomes.

While these optimizations were crucial to improving robustness, we chose not to include detailed analyses of how robustness scales with the number of training rounds and models in the main text to avoid excessive technical details. However, we now conducted three in-silico experiments to examine the effects of the number of models on the (a) prediction performance, (b) reproducibility of feature importance, and (c) consistency of the substrate prediction score. These findings are detailed in **Appendix 4**.

Re-review:

The response effectively explains that the number of models was assessed for its impact on prediction performance, robustness, and feature importance. However, it does not clearly state whether a smaller ensemble could achieve similar performance, which was the core of the reviewer's question. I also was unable to find Appendix 4.

Comment 8: The **section** on "Feature Ranking Using Machine Learning" **could be better integrated with the subsequent** "Physicochemical Signature of γ -Secretase Substrates" **section** or moved to the Methods section, as no direct results or findings are presented here

Response: We agree with the reviewer's suggestion to better integrate the "Feature ranking using machine learning" section with the "Physicochemical signature of γ -secretase substrates" section. To improve the flow, we added a transition that links the two sections by emphasizing the connection between feature ranking and the subsequent physicochemical analysis. Rather than moving this section to the *Methods*, we believe this revised structure better demonstrates the direct relevance of feature ranking to the interpretation of the physicochemical properties of substrates.

Re-review:

The comment was adequately addressed.

Comment 9: In the "Physicochemical Signature of γ -Secretase Substrates" section, the authors state, "The 10 most important CPP features comprise a cumulative feature importance of 28%, but over 100 of the top 150 features account for 100% (Fig. 1i)." **Does this imply that the last 50 features contribute minimally to γ -secretase substrate identification, while the top 100 features are sufficient?**

Response: Yes, the top 100 (more precisely top 117) features are indeed sufficient for γ -secretase substrate identification, with the last 50 (more precisely last 33) features contributing minimally. We have revised the sentence for clarity to reflect this conclusion, as shown in the following excerpt.

Re-review:

The question was adequately clarified.

Comment 10: In the "CPP and dPULearn Outperform State-of-the-Art Methods" section, the authors compare their methods with a scale-based and a deep learning-based feature engineering approach. While the deep learning-based features were derived from ProtTrans5, **more details on the scale-based embedding are expected.**

Response: We appreciate the reviewer's request for more details on the scale-based feature engineering approach. Scale-based features were derived from manually curated physicochemical properties (scales), such as hydrophobicity, charge, and secondary structure propensity. These properties were averaged across the entire transmembrane (TMD) and juxtamembrane (JMD) sequences to create a single representative value for each scale. This differs from ProtTrans5 embeddings, which automatically capture sequence features without prior knowledge of specific scales. We have revised the manuscript to clarify these distinctions, incorporating the following related comments 11 and 12.

Comment 11: The sentence, “To ensure comparability, average values were computed over the entire TMD-JMD sequences for amino acid scales and embeddings,” **should be reorganized for clarity.**

Response: We agree with the reviewer’s suggestion and have reworded the sentence for clarity, as shown in the excerpt of the revised manuscript for comment 12.

Comment 12: In the same section, the authors refer to “embeddings” in the context of feature engineering methods (scale-based, embeddings, CPP) and data expansion techniques (None, SMOTE, dPULearn). **The term “embeddings” is unclear and requires further clarification.**

Re-review (comments 10-12):

The reviewers' concerns have been well addressed in the revisions. The clarification of scale-based feature engineering, and the distinction between embeddings in different contexts effectively enhance the manuscript’s readability. Providing explicit explanations and ensuring comparability between feature engineering approaches strengthens the discussion.

Comment 13: In the section CPP and dPULearn outperform state-of-the-art methods, the authors use support vector machine models with default settings to conduct a model performance comparison. **A fairer comparison would involve using the same model architecture as in this study instead of the default support vector machine models.**

Response: We appreciate the reviewer’s suggestion and agree that using the same model architecture could offer an interesting alternative perspective. However, our primary goal in this comparison was to assess the general utility of different feature engineering approaches (scale-based, embeddings, and CPP) in a standardized machine learning setting. By using support vector machine (SVM) models with default settings, we ensured the same basic conditions where the performance differences could be attributed primarily to the feature engineering techniques rather than differences in model optimization or architecture. This allowed us to focus on how effectively each method translates raw data into meaningful features for prediction.

Additionally, SVM is a well-established baseline for binary classification, particularly suited to small datasets like ours, where overfitting is a concern. The default settings ensured that results weren’t overly influenced by hyperparameter tuning, which would vary across model architectures.

While we plan to explore the use of different models in future work, introducing them here would add unnecessary complexity and be out of the scope of this work. We believe that focusing on feature engineering with a standard SVM model provides a clear and interpretable comparison of the methods.

To reflect this discussion, we modified the current version of the manuscript, as shown in the following excerpt.

Re-review:

The response provides a justification for using SVM with default settings to isolate the impact of feature engineering techniques while avoiding confounding factors like model optimization.

However, I have concerns regarding the benchmarking of dPULearn against Elkanoto methods, particularly the absence of an external test set (see summary statement).

Comment 14: The analysis result shown in **Figure 2b** seems to be missing from the manuscript. Please provide **more details about the feature number optimization**.

Response: We appreciate the reviewer's attention to the details in Figure 2b. The feature number optimization process was conducted by evaluating different combinations of top CPP feature sets, with an increasing number of features used for model training and non-substrate identification. In total, 36 evaluations were performed (6x6 matrix), varying the number of features between 25 and 150 for both training and dPULearn (Fig. 2b). The optimization, validated by leave-one-out cross-validation, aimed to identify the best combination of top features that maximized balanced accuracy, achieving up to 90% with dPULearn when optimized. This process is detailed in the "Benchmarking CPP and dPULearn against deep learning-based embeddings" section of the *Supplementary Information* and is concisely reflected in the current version of the manuscript, as shown in the following excerpt.

Re-review:

The authors have adequately addressed this comment.

Comment 15: The fuzzy labeling strategy described in the "**Explaining Prediction Results at the Amino Acid Sequence Level**" section is confusing. It seems straightforward that the model tends to perform better on data it has seen during training than on unseen data, which is expected. **Clarification is needed.**

Response: We appreciate the reviewer's observation and agree that models generally perform better on data they have seen during training than on unseen data. However, the primary objective of fuzzy labeling is to provide reliable explanations for prediction scores of proteins not included in the initial training dataset. To achieve this, fuzzy labeling ensures that SHAP values can accurately approximate prediction scores for any protein with unknown prediction status (*i.e.*, those lacking ground-truth labels). Instead of using binary

(positive or negative) labels, fuzzy labeling applies probabilistic labels inspired by fuzzy logic, capturing varying degrees of confidence to reflect prediction uncertainty.

In this approach, machine learning models are trained over multiple rounds, including proteins whose prediction scores need to be explained but which were not part of the initial training dataset. For these proteins, labels are assigned based on their prediction scores. For example, if a protein has a prediction score of 60% and fuzzy labeling is applied over 25 training rounds, it will be labeled as positive in 15 rounds and as negative in 10 rounds. This method can be applied to proteins with unknown prediction status—in our case, proteins with unknown substrate status (*i.e.*, not reported as substrate or non-substrate in the literature). Fuzzy labeling is especially useful when prediction scores are uncertain due to high variance, as seen for low-confidence substrates or low-confidence non-substrates (**Fig. 3**).

We showcased the application of fuzzy labeling for the Alzheimer's disease-relevant protein TREM2, which is a reported substrate included in SUBLIT but was predicted as a low-confidence non-substrate ($36\pm 20\%$). This uncertainty in the substrate prediction score of TREM2 may relate to a charged lysine in the middle of its TMD—unusual for transmembrane proteins—and the absence of a well-defined TMD-C anchor, typically found in high-confidence substrates. In summary, fuzzy labeling provides a refined explanation for uncertain predictions and enhances their interpretability.

To improve clarity in the main text, we revised the section on fuzzy labeling. We first introduced the concept, emphasizing how it addresses uncertainty in predictions, and then demonstrated its practical application using the example of TREM2. This structure reflects our discussions and is shown in the following excerpt.

Re-review:

The response clarifies the purpose of fuzzy labeling and how it helps interpret uncertain predictions rather than just highlighting model performance differences on seen vs. unseen data. The explanation of probabilistic labeling and the example of TREM2 improves understandability.

Comment 16: The work related to discovering substrate subgroups based on feature impact in the "**Explaining Prediction Results at the Amino Acid Sequence Level**" section **seems redundant**, as it is straightforward that similar feature embeddings lead to similar model outputs (*i.e.*, substrate probability).

Response: We hope there was no confusion regarding the terminology used, as in the context of **Extended Data Fig. 8**, we refer to CPP features rather than embeddings. We appreciate the reviewer's observation regarding the connection between similar CPP features and similar model outputs. However, the aim of this analysis was not to confirm that similar CPP features result in similar substrate prediction scores. Instead, we sought to demonstrate how specific patterns of feature impact, as determined by the SHAP explainable AI technique, can be used to further identify groups of substrates characterized

by a common physiochemical profile, which is reflected by correlating feature impacts between substrates.

This additional layer of interpretation provides valuable insights into the underlying biological meaning of the CPP features, going beyond what can be inferred from substrate prediction scores alone. Therefore, we believe that this analysis contributes to a deeper understanding of the substrate classifications and the distinct properties that define them, offering insights beyond the expected correlation between similar features and similar substrate prediction scores.

Re-review:

The authors have provided a clear and adequate response.